# Optogenetic activation of spinal microglia triggers chronic pain in mice

**Min-Hee Yi[1], Yong U. Liu[1], Anthony D. Umpierre[1], Tingjun Chen[1], Yanlu Ying[1], Jiaying Zheng[1], Aastha Dheer[1], Dale B. Bosco[1], Hailong Dong[2], Long-Jun Wu[1,3,4]***

**1** Department of Neurology, Mayo Clinic, Rochester, Minnesota, United States of America, **2** Department of Anesthesiology & Perioperative Medicine, Xijing Hospital, Fourth Military Medical University, Xi'an, China, **3** Department of Neuroscience, Mayo Clinic, Jacksonville, Florida, United States of America, **4** Department of Immunology, Mayo Clinic, Rochester, Minnesota, United States of America

* wu.longjun@mayo.edu

**Data Availability Statement:** All relevant data are within the paper and its Supporting Information files.

**Funding:** This work was supported by the National Institutes of Health (R01NS088627;

## Abstract

Spinal microglia are highly responsive to peripheral nerve injury and are known to be a key player in pain. However, there has not been direct evidence showing that selective microglial activation in vivo is sufficient to induce chronic pain. Here, we used optogenetic approaches in microglia to address this question employing CX3CR1[creER/+]: R26[LSL-ReaChR/+] transgenic mice, in which red-activated channelrhodopsin (ReaChR) is inducibly and specifically expressed in microglia. We found that activation of ReaChR by red light in spinal microglia evoked reliable inward currents and membrane depolarization. In vivo optogenetic activation of microglial ReaChR in the spinal cord triggered chronic pain hypersensitivity in both male and female mice. In addition, activation of microglial ReaChR up-regulated neuronal c-Fos expression and enhanced C-fiber responses. Mechanistically, ReaChR activation led to a reactive microglial phenotype with increased interleukin (IL)-1β production, which is likely mediated by inflammasome activation and calcium elevation. IL-1 receptor antagonist (*IL-1ra*) was able to reverse the pain hypersensitivity and neuronal hyperactivity induced by microglial ReaChR activation. Therefore, our work demonstrates that optogenetic activation of spinal microglia is sufficient to trigger chronic pain phenotypes by increasing neuronal activity via IL-1 signaling.

## Introduction

Microglia are central nervous system (CNS)-resident immune cells that constantly survey the microenvironment through their ramified processes in the brain and spinal cord [1–3]. Following peripheral nerve injury, spinal microglia transition to reactive states in a time period correlated with pain behaviors [4,5]. In addition, many studies have shown that genetic knock-out of microglial signaling molecules or depletion of microglia partially reverse neuropathic pain [6,7], suggesting microglia as a key player in chronic pain pathogenesis. Indeed, chemogenetic activation of microglia is sufficient to trigger pain hypersensitivity [8], while chemogenetic inhibition of microglia attenuates chronic pain in rodents [9,10]. Given the complex interactions between neurons, glia, and immune cells in pain modulation, dissecting the specific role of microglia requires the continuous development of new tools.

R01NS112144; R01NS110949; R01NS110825) to L.J.W., F32NS114040 to A.D.U., and a postdoctoral fellowship from the Mayo Clinic Center for Multiple Sclerosis and Autoimmune Neurology to T.C. The funders had no role in study design, data collection and analysis, decision to publish, or preparation of the manuscript.

**Competing interests:** The authors have declared that no competing interests exist.

**Abbreviations:** ACSF, artificial cerebral spinal fluid; ChR2, channelrhodopsin-2; CNS, central nervous system; Ct, threshold cycle; DRG, dorsal root ganglion; IACUC, Institutional Animal Care and Use Committee; IHC, immunohistochemistry; IL, interleukin; IL-1ra, IL-1 receptor antagonist; i.p., intraperitoneal; i.t, intrathecal; NIH, National Institutes of Health; PFA, paraformaldehyde; ReChR, red-activated channelrhodopsin; SNT, spinal nerve transection; TM, tamoxifen; TRP, transient receptor potential.

Microglia utilize ionotropic signaling to interact with the brain microenvironment during injury and pathology [11]. Unlike neurons, which rely heavily on voltage-gated ion channels, microglia have very few voltage-gated sodium or calcium channels in vivo. Well-known ion channels mediating ionic flux in microglia include potassium channels [12], proton channels [13], transient receptor potential (TRP) channels [14], pannexin-1 [15,16], and purinergic ionotropic receptors [17]. Purinergic signaling can directly activate ionotropic P2X4 and P2X7 receptors that are highly calcium permeable [18]. In spinal microglia, P2X4 is up-regulated following nerve injury [19]. Its prolonged activation in an injury context can lead to p38-MAPK dependent BDNF release [20], pain sensitization, and the maintenance of neuropathic pain [21,22]. P2X7 receptors also promote microglial depolarization and pain responses in vivo [23]. Their prolonged activation can lead to NLRP3 inflammasome assembly and interleukin (IL)-1β release, as well as microglial activation and proliferation [24,25].

Optogenetic approaches allow manipulation of specific cell populations in real time through the activation of a light-responsive ion channel. In neurons, optogenetics has been used to drive the depolarization or hyperpolarization of selective neurons, allowing for complex circuit interrogation underlying behaviors [26]. In electrically silent astrocytes, optogenetics has also been used to dissect astrocyte function in breathing, and learning and memory [27,28]. Interestingly, channelrhodopsin-2 (ChR2) can depolarize spinal astrocytes and induce hypersensitive pain behaviors through ATP release [29]. However, so far, optogenetic tools have not been applied to study microglia. It is not yet known how experimental changes in microglia ionic conductance via optogenetic manipulation affect their cellular function or subsequent behaviors in vivo.

Using CX3CR1[creER/+]: R26[LSL-ReaChR/+] transgenic mice, we expressed the red-activated channelrhodopsin (ReChR) specifically in microglia. ReChR has improved membrane trafficking and enhanced photocurrents compared to existing red-shifted channelrhodopsins [30]. Also, red light reduces tissue scattering and absorption by chromophores, such as blood, relative to blue/green wavelengths of light. Here, we find that optogenetic stimulation of spinal cord microglia by ReChR is sufficient to produce a reactive microglial phenotype, neuronal hyperactivity, and long-lasting behavioral pain sensitization.

## Results

### Inducible expression of ReChR in microglia in CX3CR1[creER/+]: R26[LSL-ReaChR/+] transgenic mice

CX3CR1 is highly expressed by microglia in the CNS but also other cells of mononuclear origin in the periphery [31]. We generated CX3CR1[creER/+]: R26[LSL-ReaChR/+] transgenic mice to enable tamoxifen (TM) inducible Cre-dependent expression of ReChR in CX3CR1[+] cells (**Fig 1A**). To confirm ReChR expression, we stained for rhodopsin, as ReChR is a subfamily of retinylidene proteins [30]. In the spinal cord sections harvested 3 weeks post-TM injection, rhodopsin expression was observed at both the thoracic and lumbar levels of the spinal cord dorsal horn (**Fig 1Ba and 1Bb**). To characterize the cellular distribution of ReChR expression in CX3CR1[creER/+]: R26[LSL-ReaChR/+] transgenic mice, we performed co-immunofluorescence staining of Iba1 (microglia marker), GFAP (astrocyte marker), and NeuN (neuronal marker) with rhodopsin (**Fig 1C and 1D**). In mice receiving TM, the expression of rhodopsin was observed in Iba1[+] microglia but not GFAP[+] astrocytes or NeuN[+] neuronal cells in the spinal cord (**Fig 1C and 1D**). Rhodopsin expression was not observed in the spinal cord of mice that did not receive TM treatment (**S1A Fig**). In the cortex of CX3CR1[creER/+]: R26[LSL-ReaChR/+] transgenic mice, we observed the co-localization of rhodopsin and Iba1 in TM treated mice but not in mice without TM treatment (**S1B and S1C Fig**). Taken together, these results

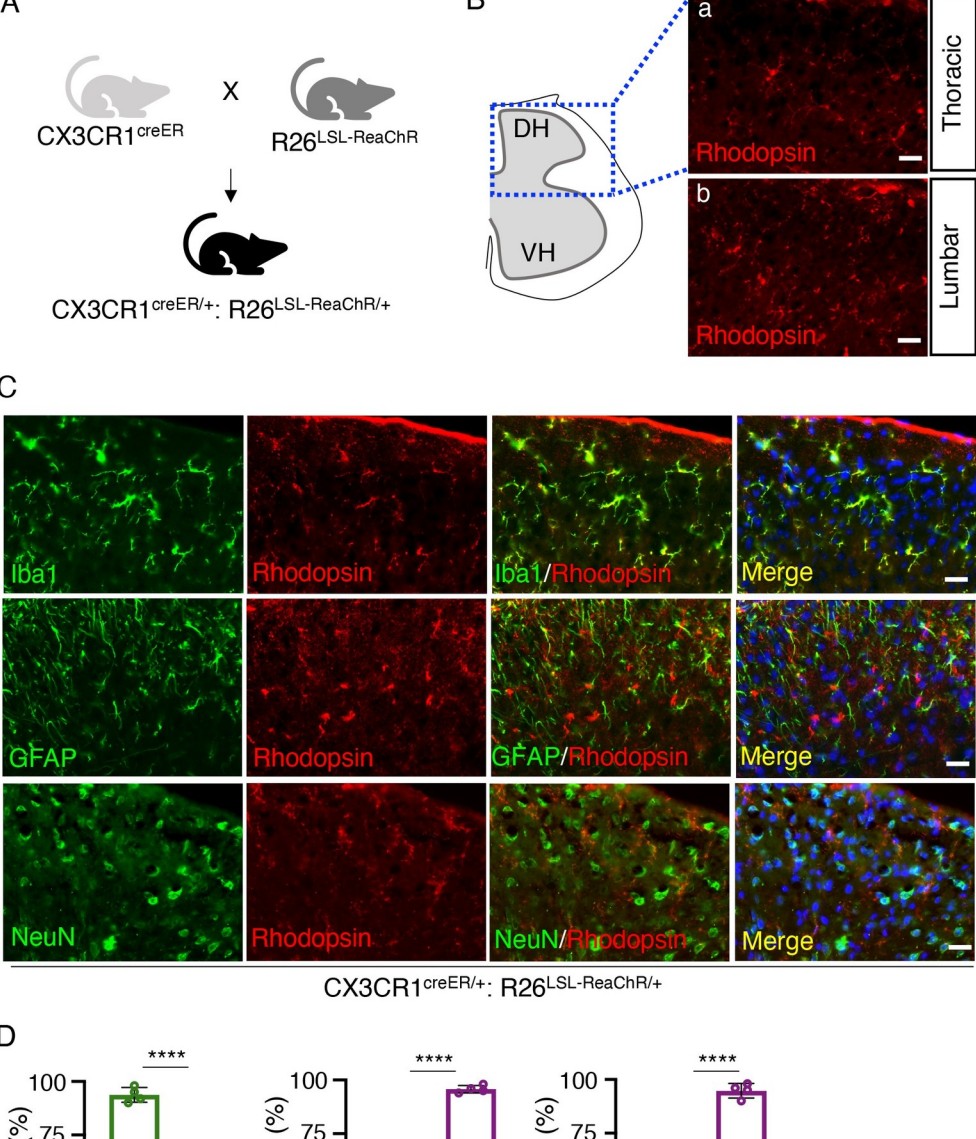

**Fig 1. CX3CR1^creER/+^: R26^LSL-ReaChR/+^ transgenic mice enable selective expression of ReaChR in spinal microglia.**
(A) Generation of CX3CR1^creER/+^: R26^LSL-ReaChR/+^ transgenic mice by crossing CX3CR1^creER/creER^ mice with R26^LSL-ReaChR^ mice. (B) Immunofluorescence images indicate rhodopsin expression in the thoracic and lumbar sections of the spinal cord. Scale bar, 40 μm. (C) Representative images of rhodopsin immunostaining (red) with either Iba1 (green), GFAP (green), or NeuN (green) in the lumbar region of the spinal dorsal horn in CX3CR1^creER/+^: R26^LSL-ReaChR/+^ mice after TM injection. Scale bar, 40 μm. $n$ = 4 mice/group. (D) Summarized data show the co-localization of rhodopsin with Iba1$^+$ cells but not with GFAP$^+$ or NeuN$^+$ cells. Data are presented as mean ± SEM. $n$ = 4 mice/group, ****$P < 0.0001$, unpaired Student $t$ test. For data plotted in graphs, see S1 Data. ReaChR, red-activated channelrhodopsin; TM, tamoxifen.

indicate that we are able to induce Cre-dependent expression of ReaChR in CX3CR1+ microglia in mouse CNS. Herein, CX3CR1$^{creER/+}$: R26$^{LSL-ReaChR/+}$ transgenic mice treated with TM referred to as "ReaChR mice," while CX3CR1$^{creER/+}$ mice undergoing similar treatment but without ReaChR expression are referred to as the "control group."

## Optogenetic stimulation of spinal microglia evokes inward currents and depolarization

To verify the functional expression of microglial ReaChR, we performed whole-cell patch-clamp recordings in microglia from acute spinal cord slices obtained from ReaChR mice (**Fig 2A**). Since mCitrine is linked with ReaChR [32], we recorded mCitrine positive microglia and examined light-induced responses. Consistent with our previous studies [33], spinal microglia exhibited small currents with a linear current–voltage relationship in response to voltage steps from −140 mV to 40 mV. We found that stimulation with red light (625 nm) via LED dramatically increased the inward currents, with little deactivation, and drove the reversal potential to around −10 mV in microglia (**Fig 2B and 2C**). We further characterized ReaChR currents by voltage ramp test with a gradual increase of light intensity (0% to 20% from 0.2 to 121.8 mW). The ReaChR currents increased in amplitude with increasing light intensity (**Fig 2D and 2E**). In addition, we performed current clamp recordings in spinal microglia to examine ReaChR-dependent changes in membrane potentials in response to different frequencies of light stimulation (persistent, 20 Hz, 10 Hz, 5 Hz, and 2 Hz). Light stimulation significantly depolarized microglia in a frequency-dependent manner with minimal desensitization, which was not observed in control group of CX3CR$^{creER/+}$ mice receiving TM (**Fig 2F**). These results indicate that ReaChR is functionally expressed in spinal microglia and mediates light-induced, nonselective channel currents and membrane depolarization.

## Optogenetic stimulation of spinal microglia induces pain hypersensitivity

To test the function of microglial ReaChR in vivo, we turned to pain behaviors in which microglia have a well-established role [6,34,35]. To this end, we optogenetically stimulated spinal microglia in vivo and monitored pain behaviors in ReaChR mice or control mice. To target spinal microglia, a ferrule was stereotaxically implanted above the vertebra of the lumbar spinal cord, and an optogenetic fiber was inserted through the ferrule for light delivery (**Fig 3A**). The optogenetic fiber was localized above the dural membrane of the dorsal horn to avoid damage to the spinal cord. Three weeks after TM injection, mice were exposed to red LED (625 nm, 45 ms light on, 5 ms light off, 20 Hz) through the optic fiber and then behavioral tests were performed (**Fig 3B**). First, we investigated the potential impacts of vertebral ferrule implantation and spinal optogenetic stimulation on motor coordination. We evaluated the motor function of each group (Sham, ferrule-implanted controls, and ferrule-implanted ReaChR) with 30-min light stimulation using gait analysis (**Fig 3C**). All groups demonstrated similar levels of locomotor ability. Using the rotarod test, we also found that there were no significant change in motor coordination across all 3 groups (**Fig 3D**). These results suggest that the procedures of ferrule implantation and optogenetic stimulation did not induce motor deficits.

Next, we investigated the effect of light stimulation with different durations (5, 15, and 30 min) on pain responses in ReaChR mice (**Fig 3E and 3F**). We found that mechanical allodynia (**Fig 3E**) and thermal hyperalgesia (**Fig 3F**) were elevated 1 h after 30-min light stimulation on the ipsilateral side, but the not contralateral side (**Fig 3E and 3F**). Additionally, elevated pain behaviors were only elicited after 30 min of stimulation and could not be achieved with 5-min or 15-min stimulation of ReaChR mice. Thus, we used 30-min light stimulation to characterize

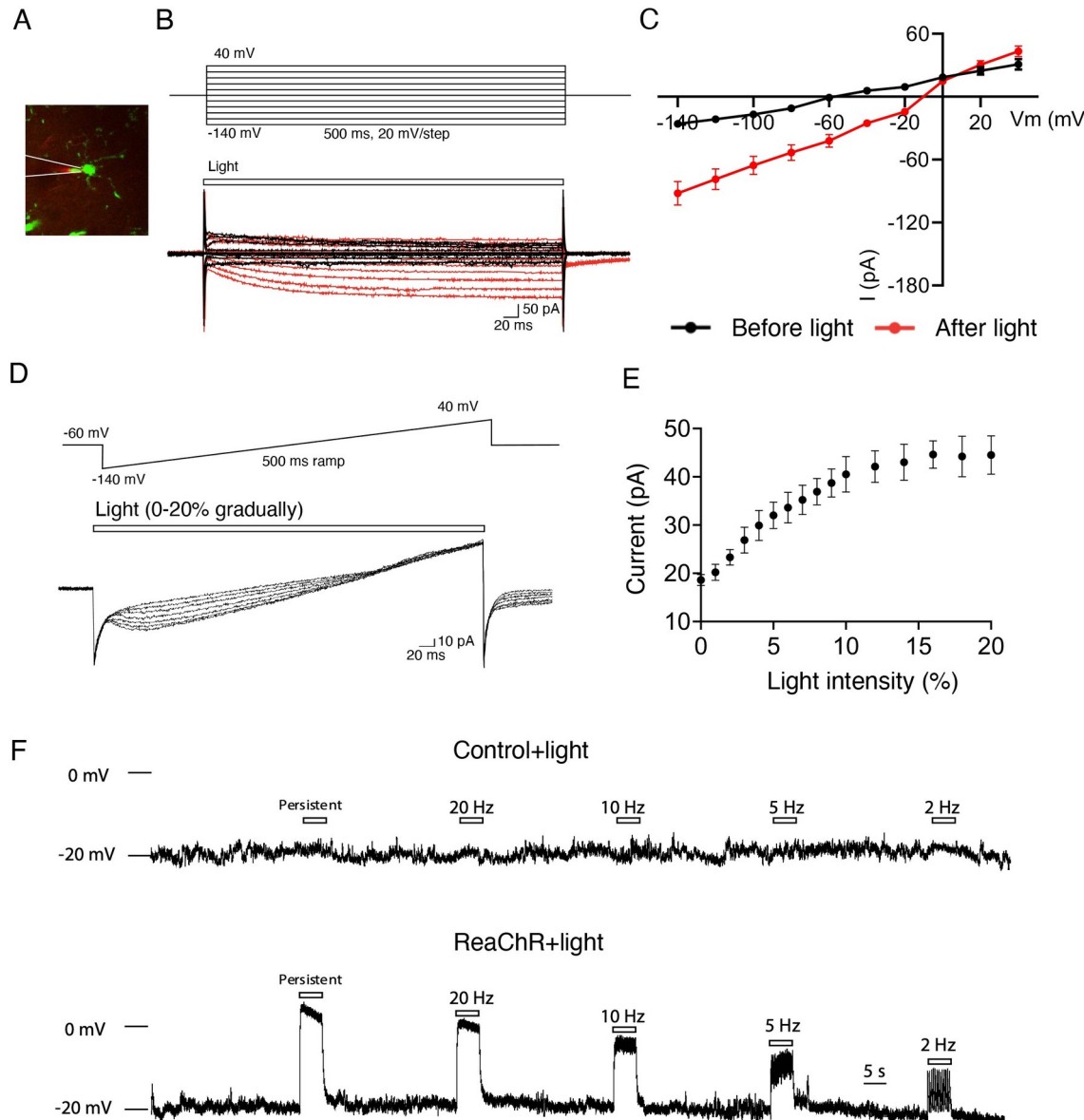

**Fig 2. Optogenetic activation of spinal microglia evokes inward currents and membrane depolarization.** (A) A representative image of mCitrine[+] microglia being recorded through whole-cell patch-clamp electrophysiology in spinal cord slices of ReaChR (CX3CR[creER/+]: R26[LSL-ReaChR/+]) mice. (B) Microglia membrane currents from baseline (before red light stimulation, black traces) or ReaChR activation (after red light stimulation [intensity 8% from 54.5 mW], red traces) in ReaChR mice. Microglia were held at −60 mV, then underwent 500-ms voltage steps from −140 to +40 mV (20 mV steps). (C) Summarized plot of current (I, [pA]) vs. holding potential (Vm, [mV]) in spinal microglia from ReaChR mice before light (black traces) and after light stimulation (red traces). (D) Voltage ramp tests from −140 to +40 mV in response to the gradual increase of light intensity (0%–20% from 0.2 to 121.8 mW). (E) A summarized graph depicts that the increase of optogenetic current in microglia correlates with the increase in light intensity. (F) The current clamp recording of microglia shows the changes in membrane potential in response to 5-s light stimulation (intensity 8% from 54.5 mW, pulse duration 45 ms) at indicated frequencies in control (CX3CR[creER/+]) mice (upper) and ReaChR mice (lower). For data plotted in graphs, see S1 Data. ReaChR, red-activated channelrhodopsin.

optogenetic activation of spinal microglia in ReaChR and control mice in the following experiments (**Fig 3G and 3H**). We found that light-induced mechanical allodynia in the ipsilateral side persisted for 7 days and recovered after 9 days (**Fig 3G**). In addition, light-induced allodynia could be reintroduced after the second optogenetic stimulation (**Fig 3G**). Interestingly,

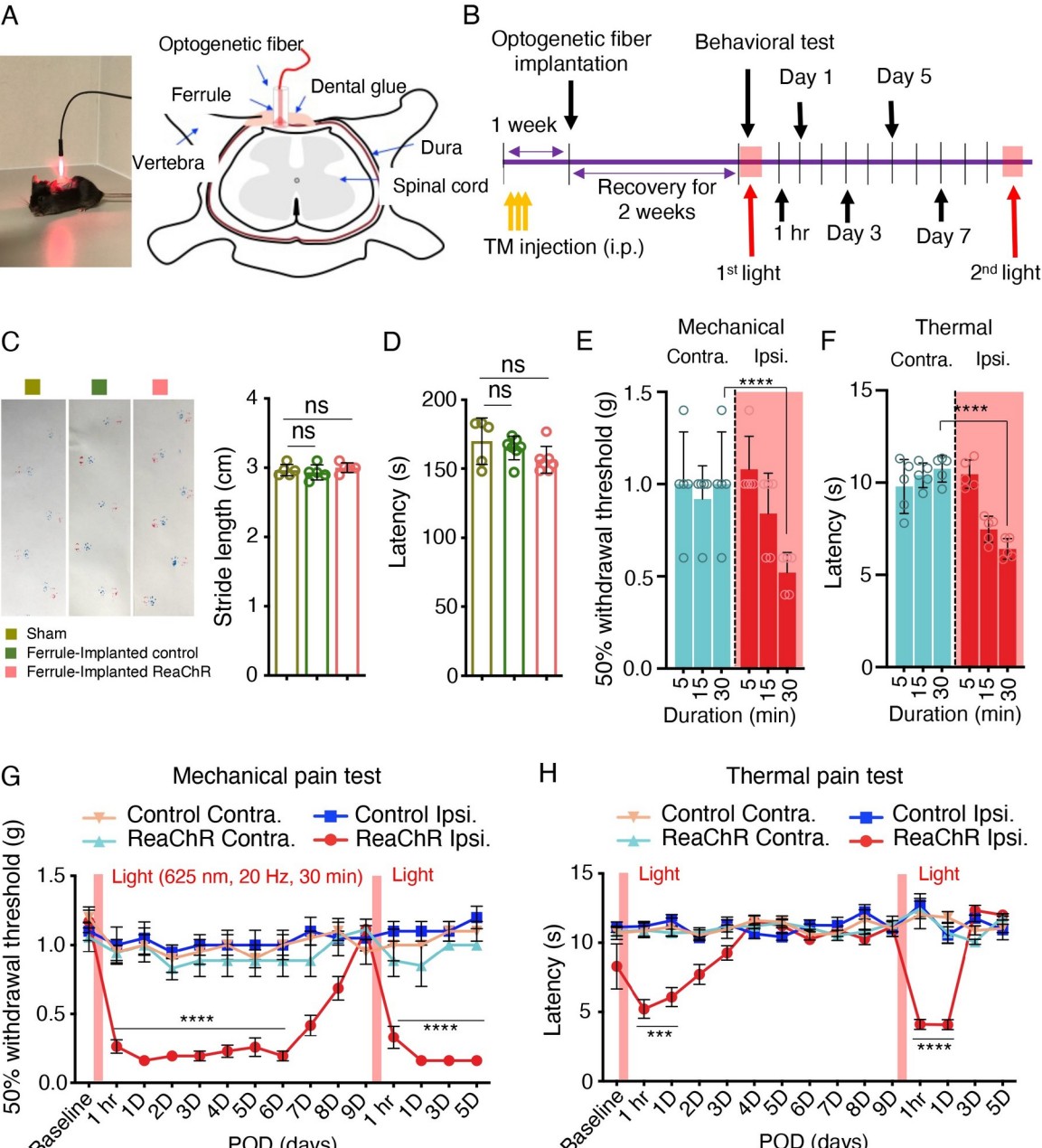

**Fig 3. Optogenetic stimulation of microglia induces mechanical allodynia and thermal hyperalgesia.** (A) Image displaying LED stimulation (red light: 625 nm, 45 ms light on, 5 ms light off, 20 Hz) in a mouse (left) and a schematic representation of the optogenetic ferrule placement region for light stimulation of mouse spinal cord (right). (B) Timeline of experimental procedures. (C) Gait analysis using the footprint test displays the stride length of mice after sham surgery, control mice after implantation of the optic ferrule (ferrule-implanted control), and ReaChR mice after implantation of the optic ferrule with ReaChR activation with 30-min duration (ferrule-implanted ReaChR). $n$ = 5 mice/group. (D) Rotarod test of motor coordination in the 3 groups described in C. $n$ = 5 mice/group. One-way ANOVA. (E, F) Mechanical allodynia (E) and thermal (F) pain hypersensitivity of ReaChR mice at 1 h after light stimulation with different durations (5, 15, and 30 min). Both contralateral (Contra.) and ipsilateral (Ipsi.) side of light stimulation were measured. $n$ = 5 mice/group. $^{****}P < 0.0001$. Two-way ANOVA with multi-comparisons. (G, H) Mechanical (G) and thermal (H) pain hypersensitivity of ReaChR mice after the first and second light stimulation for 30 min. Red bar indicates the time points of light stimulation at day 0 and day 9. Data are presented as mean ± SEM. $n$ = 7–8 mice/group. $^{***}P < 0.001$, $^{****}P < 0.0001$ Control Ipsi. vs. ReaChR Ipsi. Two-way ANOVA with multi-comparisons. For data plotted in graphs, see S1 Data. ReaChR, red-activated channelrhodopsin; TM, tamoxifen.

light-induced thermal hypersensitivity was relatively transient and only lasted for 1 day after optogenetic stimulation (**Fig 3H**). Similarly, a second round of light stimulation could still induce transient thermal hyperalgesia. We did not observe mechanical or thermal hyperalgesia on the contralateral side of ReaChR mice or in control mice (**Fig 3G and 3H**). The light-induced pain responses were not due to damage in dorsal root ganglion (DRG) neurons, as we did not observe the expression of ATF3, a well-established injury marker, after light stimulation in ReaChR or control mice (**S2A Fig**). The high expression of ATF3 was observed in ipsilateral L4 DRG but not in contralateral side after spinal nerve transection (SNT) surgery (**S2B and S2C Fig**). Together, these results indicate that optogenetic stimulation of spinal microglia is sufficient to trigger multiple forms of pain hypersensitivity in ReaChR mice.

It has been reported that microglia play a sex-dependent role in neuropathic pain [36,37]. Here, we wanted to examine whether chronic pain induced by microglial ReaChR activation is dependent on sex. To this end, we compared the effect of light stimulation on pain behaviors in male and female ReaChR mice. Interestingly, we found that ReaChR activation induced mechanical allodynia lasting for at least 5 days in both male and female mice (**S3A Fig**). However, there were noticeable differences in light-induced thermal hypersensitivity, which was transient in male mice but lasted for 5 days in female mice (**S3B Fig**).

## Optogenetic stimulation induces microglial activation in the spinal cord dorsal horn

Activation of microglia is critical for central sensitization in various pain-related conditions, including neuropathic pain after peripheral nerve injury [35,38]. Therefore, we evaluated microglial activation in the spinal cord of ReaChR mice after optogenetic stimulation. After light stimulation, the number of Iba1$^+$ microglia was significantly increased in the ipsilateral dorsal horn compared to the contralateral side in ReaChR mice (**Fig 4A and 4B**). Light stimulation did not change Iba1$^+$ microglia number in control mice. In addition, we examined the microglial proliferation after optogenetic stimulation by immunostaining for Ki67, a nuclear protein expressed in cell cycle except the resting phase [39]. We found a marked increase in proliferating microglia (Ki67$^+$Iba1$^+$ cells) at 3 days, which was reduced at 5 to 7 days in ReaChR mice after optogenetic stimulation (**Fig 4C, S4 Fig**). Next, we analyzed the morphological changes in microglia after light stimulation, which is correlated with their activation state [4,40]. Using Sholl analysis, we compared the complexity of spinal microglia at 1 h and 1 to 9 days after light stimulation in ReaChR mice and control mice. Indeed, light stimulation induced shorter processes and less complexity at 1, 3, and 5 days after light stimulation compared to controls (**Fig 4D and 4E**), but morphological changes were largely recovered at 7 to 9 days (**S5 Fig**). In addition, we examined Kv1.3, one molecular marker of microglial activation [41,42] after ReaChR stimulation. We found that there was an up-regulation of Kv1.3 in Iba1$^+$ microglia at 3 days in the ipsilateral spinal dorsal horn compared with the contralateral side after light stimulation (**Fig 4F and 4G**). The up-regulation of Kv1.3 started gradually from 30 min to 3 days, and then returned to the same level as the control mice later at 9 days (**S6 Fig**). To further characterize microglial activation, we performed western blot analysis to examine the MAPK signal transduction (Erk, p38, and JNK), which is a key mediator of microglia activation in neuropathic pain condition [19,43,44]. Indeed, we found that p-p38 and p-Erk peaked at 1 to 3 days in ipsilateral spinal dorsal horn compared control mice, but no significant difference in p-JNK was observed (**S7 Fig**). We also did not detect the activation of p-P65 in NF-$\kappa$B signaling after optogenetic activation of microglia (**S7 Fig**). Together, these results indicate that the optogenetic stimulation of ReaChR induces spinal microglia activation.

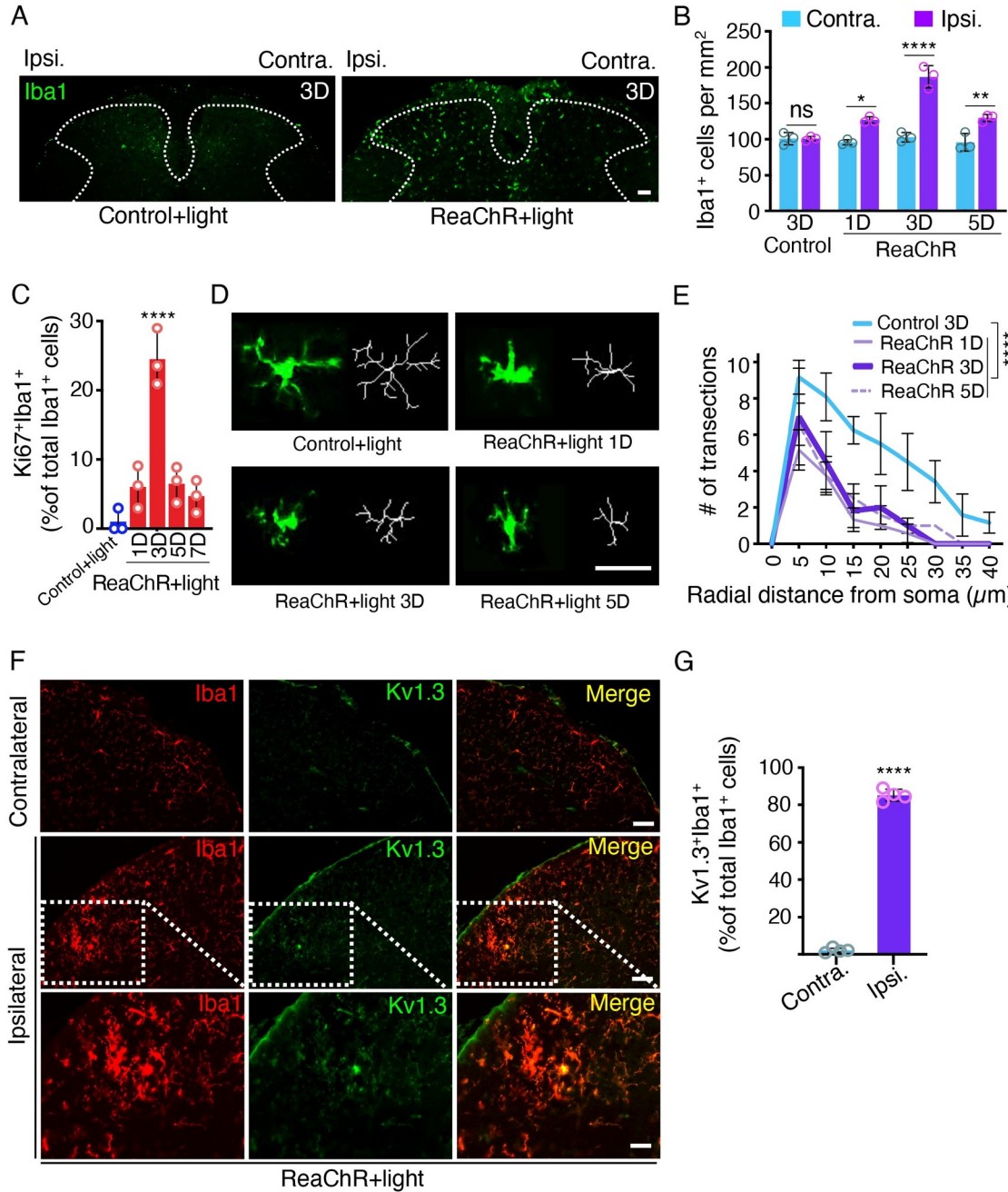

**Fig 4. Optogenetic stimulation induces microglia activation.** (A) Representative images of Iba1[+] microglia after light stimulation in control mice (left) and ReaChR mice (right). Scale bar, 40 μm. (B) Summarized data showing that the number of Iba1[+] microglia was increased in the ipsilateral dorsal horn at 1–5 days after light stimulation in ReaChR mice as compared to the contralateral side and control group. $n$ = 3 mice/group. ns. $P > 0.9999$, *$P = 0.0476$, **$P = 0.0049$, ****$P < 0.0001$ for Contra. vs. Ipsi. Two-way ANOVA with multiple comparisons. (C) A quantitative summary showing the percentage of colabelled Ki67[+] and Iba1[+] cells of the total Iba1[+] cells were significantly increased at 3 days after optogenetic stimulation compared to control group. Data are presented as mean ± SEM. $n$ = 3 mice/group, ****$P < 0.0001$. One-way ANOVA. (D) Representative single microglia images in the spinal cord using Iba1 immunostaining and after being skeletonized following optogenetic stimulation. Scale bar, 40 μm. (E) Sholl analysis indicates that optogenetic stimulation reduced the complexity of ReaChR microglia compared with the control group. $n$ = 5 mice/ group, ****$P < 0.0001$. Two-way ANOVA. (F) Representative immunostaining images showing the increased expression of Kv1.3 (green) in Iba1[+] (red) microglia at 3 days after light stimulation in the ipsilateral dorsal horn compared with the contralateral side without light stimulation in ReaChR mice. (G) Summarized data showing the co-localization of Kv1.3 with Iba1[+] cells. Data are presented as mean ± SEM, $n$ = 4 mice/group, ****$P < 0.0001$, unpaired Student $t$ test. Scale bar, 40 μm. For data plotted in graphs, see S1 Data. ReaChR, red-activated channelrhodopsin.

## Optogenetic stimulation of spinal microglia increases neuronal activity

Next, to elucidate how light-induced microglial activation translates into pain behaviors, we explored the potential effects of ReaChR stimulation on nociceptive transmission [45]. To this end, we performed in vivo recordings of C-fiber–evoked field potentials in the spinal cord dorsal horn in anesthetized mice [46] (**Fig 5A**). After obtaining stable baseline recordings for up to 60 min, we introduced 30 min of light stimulation (625 nm, 45-ms light on, 5-ms light off, 20 Hz) and assessed changes in C-fiber–evoked responses (**Fig 5B**). We found that light stimulation did not affect the C-fiber–evoked field potential in control mice, but significantly increased the responses in ReaChR mice (**Fig 5C and 5D**). The light-induced increase persisted for more than 90 min and returned to baseline over the course of 2 h. After normalization to pre-light stimulation values (baseline; 0 to 60 min before light stimulation), C-fiber–evoked field potentials were maximally increased by 298.0% ± 24.5% after light stimulation in ReaChR mice (**Fig 5C and 5D**). These results indicate that optogenetic stimulation of spinal microglia is able to facilitate nociceptive transmission in vivo.

The expression of c-Fos in spinal dorsal horn neurons is increased after noxious stimulation and associated with the development of central sensitization, neuropathic pain, and inflammatory pain [47]. To further study how microglial ReaChR activation affects neuronal activity, we examined c-Fos expression 2 h after light stimulation in the dorsal horn (**Fig 5E**). We found that c-Fos immunoreactivity was markedly up-regulated after light stimulation in the ipsilateral dorsal horn of ReaChR mice, compared with the contralateral side or control mice (**Fig 5F and 5G**). Since our results showed that short-term light stimulation (30 min) of microglial ReaChR resulted in long-lasting mechanical allodynia (5 to 7 days), we posited that microglial ReaChR activation may trigger multiple, long-term effects in neuronal activity in addition to c-Fos expression. The role of PKCα in the spinal dorsal horn plays a critical role in central sensitization and maintenance of persistent pain [48]. Here, we found a significant increase in PKCα expression mostly in NeuN$^+$ neurons at 3 days after light stimulation in ReaChR mice, compared with control mice. However, PKCα expression was not observed in Iba1$^+$ microglia after light stimulation (**Fig 5H and 5I**). Interestingly, we also observed that single injection of PKCα inhibitor peptide (10 μM, i.t), known for blocking calcium-dependent PKC translocation and function [49], prevented the light-induced chronic pain (**S8 Fig**). Thus, neuronal PKCα activation could be implicated in early time window after optogenetic activation of spinal microglia. Taken together, these results suggest that microglial optogenetic stimulation increases neuronal activity as suggested by increased neuronal c-Fos and PKCα expression/function in spinal cord dorsal horn.

## Optogenetic stimulation of spinal microglia induces pain through IL-1β signaling

During pathogenesis of neuropathic pain, activated microglia produce and release a variety of pro-inflammatory cytokines (e.g., IL-1β, TNFα, IL-6, and CCL2) and BDNF to sensitize spinal neurons [50]. To determine whether the optogenetic stimulation of microglia affects these factors in the spinal cord, we performed quantitative real-time PCR analysis. The expression of *IL-1β* but not *TNFα*, *IL-6*, and *CCL2* mRNA was significantly increased at 1 and 3 days in ReaChR mice compared to control mice. In addition, we observed *BDNF* mRNA level was enhanced at 3 days after optogenetic stimulation (**S9A–S9E Fig**). IL-1β is a pivotal pro-inflammatory mediator released from microglia [51,52]. Indeed, we found that a single intrathecal (i.t.) injection of recombinant IL-1β protein alone induces both mechanical and thermal pain responses lasting for multiple hours (**S10A and S10B Fig**). In addition, our exogenous

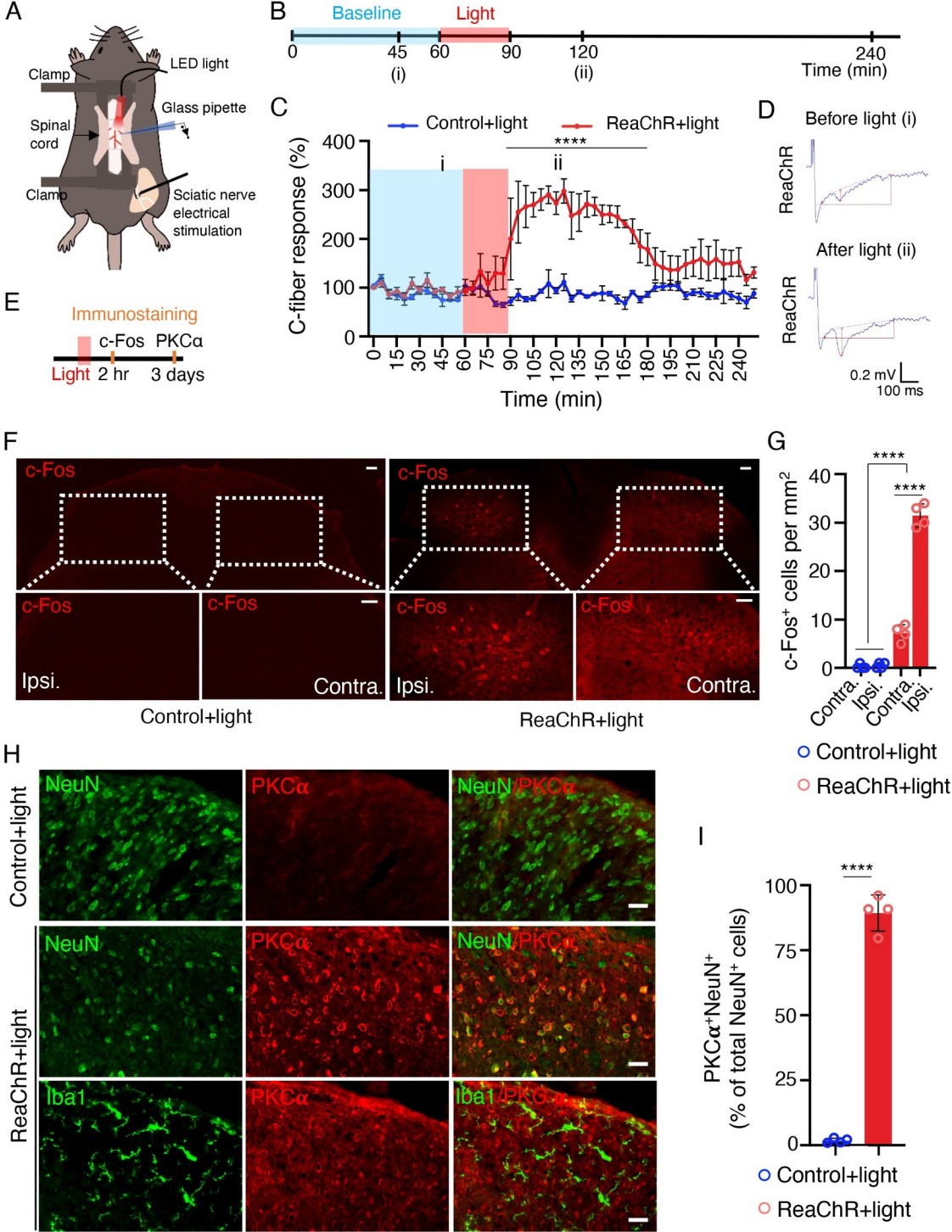

**Fig 5. Optogenetic stimulation of microglia increases neuronal activity.** (A) A schematic figure shows optogenetic stimulation and C-fiber–evoked field potential in vivo recording in mice. (B) Timeline of experimental procedures for optogenetic stimulation and in vivo electrophysiology. (C) Pooled results showing the time course of C-fiber–evoked field potentials following optogenetic stimulation of spinal microglia in ReaChR or control mice. Light stimulation enhances C-fiber responses for more than 60 min in ReaChR mice but not in control mice. C-fiber–evoked field potential was normalized to the baseline. $n$ = 4 mice/group. ****$P < 0.0001$. Two-way ANOVA with multi-comparisons. (D) Representative traces of C-fiber–evoked field potentials from ReaChR mice recorded at 45 min of baseline (i) and after light stimulation, 60 min (ii). The amplitude of C-fiber–evoked response (red vertical line) was determined with parameter extraction software WinLTP. $n$ = 5 mice/group. Scale bars, 100 ms (X) and 0.2 mV

(Y). (E) Timeline of experimental procedures for immunostaining after optogenetic stimulation. (F) Representative immunostaining images showing the c-Fos expression in ipsilateral side (Ipsi.) of spinal cord in ReaChR mice at 2 h after light stimulation but not in contralateral side (Contra.) or in control mice. The insets show the magnified images of boxed areas. Scale bar, 40 μm. (G) Summarized data showing the c-Fos$^+$ cells. $n$ = 4 mice/group, $^{****}P < 0.0001$. Two-way ANOVA with multi-comparisons. (H) Immunofluorescence images of PKCα (red) with either Iba1 (green) or NeuN (green) in the spinal cord in ReaChR or control mice after light stimulation. PKCα was not detected in the control group. At 3 days after light stimulation in ReaChR mice, PKCα was co-localized with NeuN$^+$ neurons but not with Iba1$^+$ microglia in the spinal dorsal horn. (I) Summarized data showing the co-localization of PKCα with NeuN$^+$ cells. $n$ = 4 mice/group, $^{****}P < 0.0001$. Unpaired Student $t$ test. Data are presented as mean ± SEM. For data plotted in graphs, see S1 Data. ReaChR, red-activated channelrhodopsin.

application IL-1β protein was able to up-regulate endogenous IL-1β in microglia (**S10C and S10D Fig**), suggesting an amplified response for microglial IL-1β signaling.

Here, we first tested whether light stimulation is able to induce IL-1β expression in microglia in vivo. Indeed, we observed that IL-1β was largely induced and selectively expressed in Iba1$^+$ microglia in the ipsilateral dorsal horn at 1 day after light stimulation. However, there was very little IL-1β expression in control mice (**Fig 6A and 6B**). Next, we wanted to examine the mechanism underlying IL-1β production and secretion after optogenetic stimulation of microglia. To this end, we first considered NLRP3 inflammasome activation, which is known to induce caspase-1 mediated IL-1β secretion [53]. We performed western blot analysis and observed that protein expression of NLRP3 inflammasome components and caspase-1 was markedly up-regulated in ReaChR mice after optogenetic stimulation compared with control group (**Fig 6C and 6D**). Consistently, in vitro experiment using cultured primary microglia confirmed that optogenetic activation of microglial ReaChR increases IL-1β release detected by ELISA. Light stimulation induced IL-1β release in the culture media from ReaChR expressing microglia at 1, 6, or 24 h compared with the control group in vitro (**Fig 6E–6G**). However, we did not observe significant BDNF release after light stimulation (**S9F Fig**).

Ca$^{2+}$ signaling is essential for NLRP3 inflammasome activation [54]. Channelrhodopsins are known as Ca$^{2+}$ permeable and light-activated ion channels for triggering Ca$^{2+}$ influx [55]. Therefore, we further examined the potential role of Ca$^{2+}$ in promoting ReaChR-induced release of IL-1β. To this end, we used EGTA (5 mM) to chelate extracellular Ca$^{2+}$ during light stimulation in cultured microglia from ReaChR mice. We observed that the released IL-1β in the culture medium was completely inhibited by chelating extracellular Ca$^{2+}$ (**Fig 6H**). Thus, Ca$^{2+}$ elevation by ReaChR activation is implicated in promoting the release of IL-1β from microglia. Recent studies showed that pannexin-1 activation can drive the release of IL-1β from microglia in inflammatory pain [15,16]. To investigate the involvement of pannexin-1 in microglial ReaChR activation-induced chronic pain, we used probenecid (100 mg/kg), a broad-spectrum pannexin-1 blocker [56]. However, single systemic injection of probenecid did not prevent the progressive reduction in mechanical threshold or reversed the established mechanical allodynia after optogenetic activation of spinal microglia (**S11 Fig**).

To delineate whether IL-1β is the causal factor that mediated the increase of synaptic transmission and pain, we tested the effect of an IL-1 receptor antagonist (*IL-1ra*) to inhibit IL-1β signaling. To this end, we recorded C-fiber–evoked field potentials using in vivo recordings in the spinal cord of anesthetized mice. Three groups of mice were used: ReaChR+vehicle, ReaChR+IL-1ra, and Control+IL-1ra. Vehicle or IL-1ra (10 μL, 50 ng/ml) was applied after 30 min of baseline recording. IL-1ra application alone did not change the baseline C-fiber–evoked responses in any group. We then gave light stimulation at 30 min after the administration of IL-1ra (**Fig 7A**). After normalization to pre-light stimulation values, we consistently found increased C-fiber–evoked field potentials in the ReaChR group by light stimulation (**Fig 7B**). However, IL-1ra administration prevented light-induced increase in C-fiber–evoked field potentials (**Fig 7B**). Thus, these results demonstrate that increased nociceptive transmission

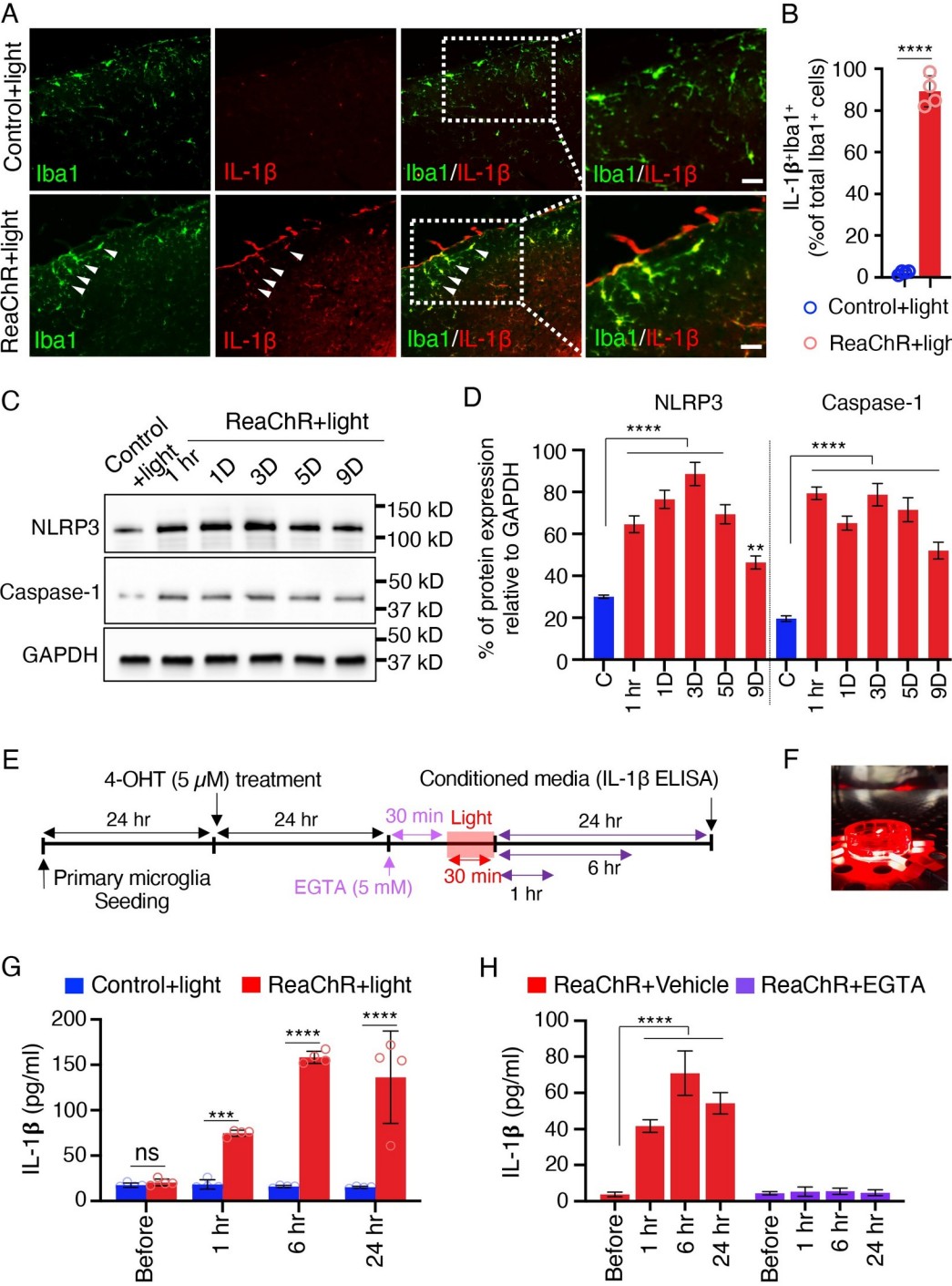

**Fig 6. Optogenetic stimulation of microglia increases IL-1β production and release.** (A) Representative immunostaining images showing IL-1β (red) up-regulation in Iba1$^+$ (green) cells in the spinal cord of ReaChR mice at 1 day after light stimulation but not in the control group. Scale bar, 40 μm. (B) Summarized data showing the co-localization of IL-1β with Iba1$^+$ cells. $n$ = 4 mice/group, $^{****}P < 0.0001$. Unpaired Student $t$ test. (C, D) Representative western blot images and quantification data showing NLRP3, Caspase-1, and IL-1**β** expression in the L4-5 level of the dorsal horn after optogenetic stimulation. Data are presented as mean ± SEM. $n$ = 4–6 mice/group, GAPDH was used as internal control. $^{****}P < 0.0001$. One-way ANOVA with multi-comparisons. Uncropped western blot images were included in the Supporting information file. (E) Timeline of experimental procedures for ReaChR induction, optogenetic stimulation, and IL-1β ELISA from cultured microglia. (F) Image showing light stimulation of primary ReaChR microglia in culture. (G) Pooled ELISA results indicating the released IL-1β in culture media of primary microglia following optogenetic stimulation. Light stimulation increases IL-1β

release at 1, 6, or 24 h from ReAChR expressing microglia but not from control microglia culture. $^{***}P < 0.001$, $^{****}P < 0.0001$. Two-way ANOVA with multi-comparisons. (H) Pooled ELISA results showing no significant release of IL-1β in culture media of primary microglia treated with EGTA (5 mM) following optogenetic stimulation. Results indicate that EGTA prevented IL-1β release at 1, 6, or 24 h from ReAChR expressing microglia after light stimulation but not from the vehicle treatment group. $^{****}P < 0.0001$. Two-way ANOVA with multi-comparisons. For data plotted in graphs, see S1 Data. IL, interleukin; ReAChR, red-activated channelrhodopsin.

after microglia ReAChR activation is mediated by IL-1β signaling. We further investigated the effects of IL-1ra administration on light-induced pain hypersensitivity. Animals were intrathecally injected with IL-1ra (10 μL, 50 ng/ml) 30 min before light stimulation (Fig 7C and 7D). We found that IL-1ra administration significantly alleviated light-induced mechanical allodynia (Fig 7C) and thermal hyperalgesia (Fig 7D) compared with the vehicle injected group. Consistently, IL-1ra administration also inhibited neuronal expression of PKCα in spinal neurons by light stimulation (S12 Fig). In sum, these findings suggest that IL-1β from optogenetically activated microglia mediates synaptic potentiation and pain hypersensitivity.

## Discussion

In this study, we used a microglial optogenetic approach as a novel tool to induce microglial activation and examined its function in pain behaviors. Taking advantage of CX3CR$^{creER/+}$: R26$^{LSL-ReAChR/+}$ transgenic mice, we were able to controllably and specifically activate microglial ReAChR to induce the depolarization of spinal microglia. We found short-term activation of microglia via direct optogenetic stimulation leads to long-lasting changes in neuronal activity and chronic pain behaviors. Mechanistically, optogenetic stimulation of microglia leads to IL-1β production that increases C-fiber–evoked responses, which could mediate the chronic pain hypersensitivity (Fig 7E). Our results demonstrate that optogenetic activation of ReAChR in spinal microglia is sufficient to trigger chronic pain behaviors, indicating that microglial optogenetic approaches represent a unique and controlled way to selectively study microglial function in awake mice.

Optogenetics has been mainly used to interrogate the neuronal circuits underlying various brain functions [57]. Recent studies have also applied optogenetics to manipulate glial activity, particularly astrocytes, in the normal and diseased brain [27–29,58]. Despite the increasing interest in microglial function in the CNS, optogenetic tools have not been used to study microglia due to the lack of effective viral tools for microglial research [59,60]. To this end, we first generated CX3CR$^{creER/+}$: R26$^{LSL-ReAChR/+}$ transgenic mice, which inducibly express ReAChR in CNS microglia and validated its function. In addition, ReAChR-activated spinal microglia exhibited morphological signs of activation alongside IL-1β expression. Most importantly, we found that short-term optogenetic stimulation of spinal microglia is sufficient to trigger long-lasting pain hypersensitivity. Our current study is exciting in several regards. First, this is the first genetic mouse model to manipulate microglia functions in vivo using optogenetic approaches. Second, our results suggest that transient optogenetic activation of spinal microglia is sufficient to trigger chronic pain behaviors. Third, we showed that microglial IL-1β via optogenetic activation is a critical mediator for synaptic potentiation and pain hypersensitivity. Therefore, this study provides proof of principle that optogenetics is a viable tool for understanding microglia function in chronic pain. We are aware of several caveats in this study. (1) CX3CR1 is expressed in some peripheral immune cells and resident CNS macrophages in addition to microglia. Although our TM injection paradigm excludes mostly the peripheral immune cells, we cannot exclude the possibility of ReAChR expression in resident perivascular or meningeal macrophages. (2) Recent reports suggest leaky expression with CX3CR1$^{CreER}$ mice, depending on the length of the STOP cassette in this loxP system [61].

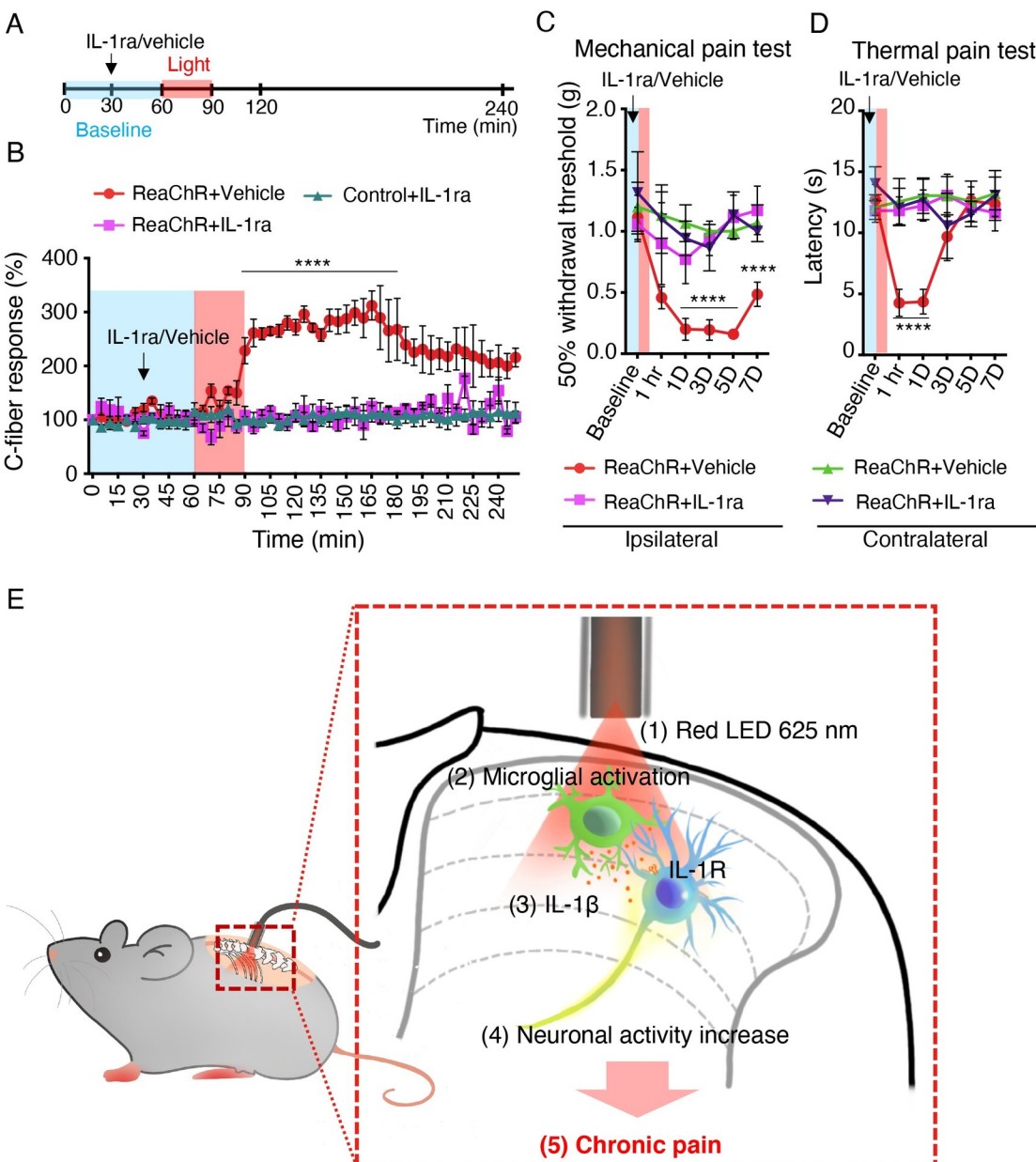

**Fig 7. Inhibition of IL-1β alleviates microglial ReaChR-induced pain hypersensitivity.** (A) Timeline of experimental procedures for IL-1ra administration and light stimulation during in vivo recordings. (B) The time course of C-fiber–evoked field potentials before and after optogenetic stimulation of spinal microglia in each group (ReaChR+vehicle, ReaChR+IL-1ra, and control+IL-1ra). Vehicle and IL-1ra (10 μL, 50 ng/mL) was topically administered after 30-min baseline recording and subsequently light stimulation was applied at 60 min. Black arrows and red bar indicate the time points of IL-1ra treatment and light stimulation, respectively (A, B). $n$ = 4–5 mice/group. $^{****}P < 0.0001$ vehicle vs. IL-1ra. Two-way ANOVA with multi-comparisons. (C, D) Light-induced mechanical (C) and thermal (D) pain hypersensitivity in ipsilateral or contralateral side of ReaChR mice after treatment of IL-1ra or vehicle. Data represented as mean ± SEM, $n$ = 5–8 mice/group. $^{****}P < 0.0001$ vehicle vs. IL-1ra. Two-way ANOVA with multi-comparisons. For data plotted in graphs, see S1 Data. (E) A schematic diagram illustrates that optogenetic stimulation of spinal microglia triggers chronic pain in ReaChR mice and highlights the possible mechanism involved. Red light stimulation (1) on the spinal dorsal horn induces microglial activation (2), which releases IL-1β (3) and increases neuronal hyperactivity (4), leading to chronic pain hypersensitivity (5). IL, interleukin; IL-1ra, IL-1 receptor antagonist; ReaChR, red-activated channelrhodopsin.

Nevertheless, our control and experimental expression data sets do not indicate that issue. Also, we confirmed the specific expression of ReaChR in Iba1$^+$ microglia after TM injection in the CNS. To circumvent these issues of CX3CR1$^{CreER}$ mice, future studies should use the newly available microglia specific cre lines including TMEM119$^{CreER}$ [62] and Hexb$^{CreER}$ mice [63].

The advantage of optogenetics is that it targets cells with high spatiotemporal precision [64]. Although microglia cannot fire action potentials, they respond dramatically to ion flux due to the high membrane resistance [65]. Indeed, we found that activation of microglial ReaChR reliably induces membrane depolarization in microglia. To further validate this new optogenetic tool in microglia in vivo, we chose to study pain behaviors, because the role of microglia in pain has been well established [6, 35]. For instance, spinal microglia are known to participate in synaptic plasticity [46], central sensitization [66], and neuropathic pain [22]. Furthermore, Nasu–Hakola patients having TREM2 mutations in microglia show pain symptoms [67]. Consistently, our results demonstrate that optogenetic activation of spinal microglia is sufficient to induce reversible pain hypersensitivity. These results suggest an intriguing possibility for "microgliogenic" pain that originates from microglial dysfunction in the CNS. We used ReaChR activated by red light that is more suitable for spinal activation without penetration of optic fiber into spinal parenchyma [30]. Thus, ReaChR is advantageous than ChR2 (activated by blue light) in studying spinal mechanism of pain. We also found that ReaChR is widely expressed in different brain regions such as the hippocampus and cortex. Therefore, CX3CR1$^{creER/+}$: R26$^{LSL-ReaChR/+}$ transgenic mice will be a useful tool to study supraspinal microglia in brain function, such as pain-related comorbidity [68], memory [69], and epilepsy [70].

Our optogenetic approach to alter microglial function is based on the known literature of microglial ion channel conductance and modeling their role in pain condition. A variety of ion channels including K$^+$ channels, Cl$^-$ channels, TRP channels, proton channels, and ligand-gated ion channels such as P2X4 and P2X7 play important functions in cell proliferation, production of cytokines and cytotoxic substances, morphological changes, and the migration of microglial cells [71–73]. Here, we artificially express ReaChR in microglia and observe a light-induced inward current, consistent with ReaChR as a nonselective cation channel [30]. We found that microglial ReaChR activation in vivo can increase synaptic transmission, neuronal excitability (c-Fos and PKCα expression), and long-lasting pain behaviors. Thus, these results suggest that opening of nonselective cation channel like ReaChR in microglia is sufficient to enhance neuronal activity and pain. In line with this idea, there are several nonselective cation ion channels in microglia, including P2X4, P2X7, and TRPM2, that are reported to be critical for neuropathic pain induction [6]. Particularly, our results by optogenetic activation of ReaChR in spinal microglia are reminiscent of a previous study in which the transplantation of P2X4-activated microglia induces pain behaviors [19]. Therefore, ReaChR likely mimics other endogenous, nonselective cation channels in microglia and is a unique optogenetic tool to understand microglia function.

How ReaChR activation in the spinal microglia leads to neuronal hyperactivity and chronic pain? There are 2 potential consequences after ReaChR activation, including membrane depolarization and Ca$^{2+}$ elevation. Currently, we know little about the function of membrane potential alterations in microglia function. It has been reported that K channel activation and hyperpolarization is critical for microglial motility [33,74]. In addition, spinal microglia reactivity is correlated with K channel activation in neuropathic pain [75,76]. Therefore, these results suggest that membrane hyperpolarization (associated with K channels) instead of depolarization in microglia may activate microglia, likely due to the increase of Ca$^{2+}$ driving force. Indeed, a recent report demonstrates that inhibition of K channel in microglia decreases P2X4-dependent Ca$^{2+}$ elevation [12]. Therefore, we believe that ReaChR-mediated Ca$^{2+}$ elevation other than membrane depolarization might be more important in light-induced microglial activation and subsequent neuronal PKCα activation and chronic pain behaviors.

Although we did not test the role of neuronal PKCα in maintaining chronic pain, previous studies have suggested that PKCα activation leads to GluR2 internalization in dorsal horn neurons, which increases $Ca^{2+}$ permeability and promotes chronic pain [48].

Microglial $Ca^{2+}$ activity was not well studied in vivo but was suggested to be critical for microglial functions such as motility and cytokine release [77–79]. Although we did not directly detect intracellular $Ca^{2+}$ increase in microglia after optogenetic ReaChR stimulation, the light-induced inward current likely contains $Ca^{2+}$ influx through ReaChR. ChR2 is known as $Ca^{2+}$ permeable and light-activated ion channel for triggering $Ca^{2+}$ influx itself [55]. Considering $Ca^{2+}$ increase is a common signaling pathway in potential release and expression of gliotransmitters, cytokines, and chemokines [80], ReaChR-induced $Ca^{2+}$ might be critical in microglial regulation of neuronal activity. In addition, ChR2-mediated extracellular $K^+$ elevations may also affect the excitability of neurons [81], while ChR2-mediated intracellular $K^+$ reduction may be critical for pro-inflammatory cytokines like IL-1β [82]. We demonstrate that IL-1β is the downstream molecule after ReaChR activation for the following reasons: (1) ReaChR activation induces IL-1β up-regulation in microglia; (2) microglial ReaChR activation increases the expression of NLPR3 and caspase-1; (3) chelating extracellular $Ca^{2+}$ abolishes ReaChR-induced IL-1β release; (4) inhibition of IL-1β by IL-1ra ameliorates microglial ReaChR-dependent synaptic potentiation in the spinal cord; and (5) IL-1ra also inhibits light-induced neuronal activation and chronic pain. Thus, ReaChR-dependent $Ca^{2+}$ elevation and decreased intracellular $K^+$ could facilitate the production and release of IL-1β which potentiates synaptic transmission and initiate chronic pain hypersensitivity. Interestingly, activation of P2X4 in spinal microglia coupled with BDNF production is critical for neuropathic pain [20]. In addition, optogenetic activation of spinal astrocytes induced pain hypersensitivity via ATP release [29]. Therefore, future studies are needed to test whether other factors such as BDNF and ATP mediate microglial ReaChR-dependent chronic pain. Since ReaChR is an exogenous cation channel, $Ca^{2+}$ signaling induced by ReaChR activation may not represent the physiological microglia function. In addition, optogenetic stimulation of microglia is probably unnaturally strong. Therefore, there were limitations to use optogenetic approaches for studying microglia function in natural chronic pain conditions. Development of new optogenetic tools other than ChR2, such as stim1 channel [83], which is highly expressed in microglia, might be useful in dissecting microglia function in neuronal circuits and behaviors.

As resident immune response cells, microglia are well known to employ their function in a variety of neurological diseases, such as bacterial meningitis, ischemic stroke, epilepsy, Alzheimer disease, Parkinson disease, multiple sclerosis [70,84,85]. By understanding microglia in the normal and diseased brain, researchers are developing tools to manipulate microglia with high spatial and temporal resolution. The new optogenetic approach using ReaChR offers such an opportunity and is unique in studying microglial $Ca^{2+}$ signaling in vivo. This novel tool will advance our ability to understand how microglia respond to the microenvironment as well as regulate neuronal activity and subsequent behaviors. Interestingly, the safety of using viral vectors and opsins in optogenetics was demonstrated in preclinical studies [86, 87]. Although the idea is far-fetched, optogenetic manipulation of microglia might be theoretically explored as therapeutic target in pain management and in other microglia-related neurological disorders.

## Methods

### Ethics statement

All experimental protocols were conducted according to the National Institutes of Health (NIH) guidelines for animal research and approved by the Institutional Animal Care and Use Committee (IACUC) at Mayo Clinic (IACUC protocol # A00002735-17).

## Animals

C57BL/6J, R26$^{LSL-ReaChR}$ (026294) [32], and CX3CR1$^{CreER/CreER}$ mice (021160) [88] were purchased from Jackson Laboratory (Bar Harbor, Maine, United States of America). CX3CR1$^{CreER/CreER}$ mice were crossed with R26$^{LSL-ReaChR}$ mice to obtain CX3CR1$^{creER/+}$:R26$^{LSL-ReaChR/+}$ offspring used in experiments. Male mice (7 to 12 weeks old) were used throughout the study unless the use of female mice was specifically indicated. These mice were assigned to an experimental group randomly within a litter. Experimenters were blind to drug treatments.

## Cre-dependent ReaChR expression

To induce ReaChR expression in microglia, 150 mg/kg TM (T5648, MilliporeSigma, Burlington, Massachusetts, USA) in corn oil (20 mg/mL) was administered via intraperitoneal (i.p.) injection once daily for 3 days.

## Optogenetic ferrule implantation and light stimulation

Mice were anesthetized by 2% isoflurane anesthesia on a stereotaxic apparatus. An incision was made along the skin on the back of the animal, and then connective tissue and muscles were removed, exposing the bones of the lumbar spinal cord region. The spinal cord was immobilized in place using a metal bar along the vertebrae. A small laminectomy was made through the bone using a high-speed dental drill. A ceramic ferrule (Thorlabs, CF230-10, ø2.5 mm, 10.5 mm, New Jersey, USA) securing a length of optic fiber (Thorlabs, FT200UMT, ø200 μm) was inserted into the hollow space of the ferrule. By making the ferrule and optic fiber equal in length, we can assess the site of stimulation without damaging the spinal cord tissue. Therefore, the optic fiber tip was exposed (within 0.3 mm), reaching the bone surface line for light delivery, in the epidural space. The ferrule was secured in place using dental glue (Ivoclar Vivadent, Tetric EvoFlow, Schaan, LIE). In this procedure, we ensured the dura membrane was left intact, and the spinal cord tissue was unscathed by the procedure. Mice were provided 2 weeks to recover from the surgery.

To activate ReaChR in microglia, light stimulation was delivered through the optic fiber implanted with the ferrule. We used 625-nm, red LED light (20 Hz: 45-ms light on, 5-ms light off, Thorlabs, M625F2, 13.2 mW) to activate the ReaChR. As a control, CX3CR1$^{creER/+}$ mice with TM injection (i.p.) were also given identical stimulation patterns and protocols.

## Behavioral measurement

Mechanical allodynia was assessed by measuring the paw withdraw threshold using a set of von Frey filaments (0.04 to 2 g; North Coast Medical, California, USA). Mice were placed on an elevated metal grid. The filament was applied to the plantar surface from the bottom at a vertical angle for up to 3 s. Fifty percent of withdrawal threshold values were determined using the up–down method [89].

Thermal hyperalgesia was assessed by measuring the paw withdrawal latency in response to radiant heat stimuli. Mice were placed in elevated chambers with a plexiglass floor and allowed to habituate for 20 min. The radiant heat source (IITC Life Science, California, USA) was applied to the center of the plantar surface of the hind paw 4 times with 3-min intervals. The average withdrawal latency of the 4 trials was recorded as the response latency.

Rotarod tests were performed using a four-lane Rotarod apparatus (Med Associates, Vermont, USA). The rotarod speed started at 4 RPM and uniformly accelerated over a 5-min

period to 40 RPM. Each mouse was tested 3 times in a day with a 15-min interval. The average daily latency per mouse from the 3 trials was used for analysis.

Gait analysis was performed using the footprint test. The forepaws and hindpaws of the mice were coated with red and blue nontoxic paints, respectively. The mice were trained to walk along a covered runway (50-cm length, 10-cm width, with 10-cm walls). All the mice were given 3 runs per test using a fresh sheet of white paper. Hind limb stride length was calculated using the average length between 4 sequential steps. This value was averaged among the 3 daily trials per mouse.

## Fluorescent immunostaining

Mice were deeply anesthetized with isoflurane (5% in $O_2$) and perfused transcardially with 20-ml PBS followed by 20 ml of cold 4% paraformaldehyde (PFA) in PBS. The spinal cord was removed and postfixed by submersion in the same 4% PFA solution for 4 to 6 h at 4˚C. The samples were then transferred to a 30% sucrose in PBS solution overnight for cryoprotection. Sample sections (15 µm in thickness for immunohistochemistry [IHC] or 30-µm thickness for Sholl analysis) were prepared on gelatin-coated glass slides using a cryostat (Leica, Hesse, DEU). The sections were blocked with 5% goat or donkey serum and 0.3% Triton X-100 (Sigma Aldrich, Missouri, USA) in PBS buffer for 60 min, and then incubated overnight at 4˚C with primary antibody: rabbit-anti-rhodopsin (1:200, Abcam Cat# ab5417, RRID: AB_304874, Cambridge, GBR), rabbit-anti-Iba1 (1:500, Abcam Cat# ab178847, RRID: AB_2832244), goat-anti-Iba1 (1:500, Abcam Cat# ab5076, RRID:AB_2224402), rabbit-anti-NeuN (1:500, Abcam Cat# ab104225, RRID:AB_10711153), rabbit-anti-GFAP (1:500, CST Cat#12389s, RRID:AB_2631098), mouse-anti-PKCα (1:200, Abcam Cat# ab31, RRID: AB_303507), mouse-anti-IL-1β (1:400, Cell Signaling Technology Cat# 12242, RRID: AB_2715503, Massachusetts, USA), mouse-anti-Kv1.3 (NeuroMab Cat# 73–009, RRID: AB_10673575, California, USA), rabbit-anti-c-Fos (Cell Signaling Technology Cat# 2250, RRID:AB_2247211) rabbit-anti-Ki67 (Abcam Cat# ab15580, RRID:AB_443209), and rabbit-anti-ATF3 (Santa Cruz Biotechnology Cat# sc-188, RRID:AB_2258513, Texas, USA). The sections were then washed and incubated for 60 min at room temperature with an appropriate secondary antibody: goat anti-rat (1:500, Thermo Fisher Scientific Cat# A-11006, RRID: AB_2534074, Massachusetts, USA), goat anti-rabbit (1:500, Thermo Fisher Scientific Cat# A-11035, RRID:AB_2534093), donkey anti-goat (1:500, Thermo Fisher Scientific Cat# A-11058, RRID:AB_2534105), or goat anti-mouse (1:500, Thermo Fisher Scientific Cat# A-11029, RRID:AB_2534088). The sections were mounted with Fluoromount-G (SouthernBiotech, Alabama, USA), and fluorescent images were obtained with the EVOS FL Imaging System.

## Sholl analysis

Fixed tissue (30 µm) was used to acquire Z-stack images of microglia (3-µm step size) using a 40× objective and a confocal microscope (LSM510, Zeiss, Oberkochen, DEU). Consecutive Z-stack images were converted to a maximum intensity projection image using Fiji software. Over the maximum intensity projection image, concentric circles were drawn (concentric circles plugin, fiji), centered on the soma, beginning at a 0.5-µm radius and increasing in radius by 0.1-µm steps. Sholl analysis was manually performed for each cell by counting the number of intersections between microglia branches along each concentric circle to create a Sholl plot. Additional measures to characterize each cell included the process branch number, process length, and cell soma area.

## *In vivo* extracellular recordings in the dorsal horn

Mice were anesthetized with urethane (Sigma Aldrich, catalogue #U2500 1.5 g/kg, i.p.). A T12-L1 laminectomy was performed to expose the lumbar enlargement of the spinal cord. Under a surgical microscope, the dura was carefully removed, and the left sciatic nerve was gently dissected free for electrical stimulation using a bipolar platinum hook electrode. Stabilizing clamps were affixed to the spinal column, head, and tail (model STS-A; Narishige, New York, USA) to minimize movement-associated artifacts during recording. A small recording well was formed on the dorsal surface of the spinal cord for drug application (IL-1ra, 50 ng). Single, test stimuli (0.5-ms duration, 1-min intervals, at C-fiber intensity) were delivered to the sciatic nerve. C-fiber–evoked fEPSP were recorded from the dorsal horn of the mouse spinal cord using a glass microelectrode filled with 0.5-M sodium acetate (impedance 0.5 to 1 M$\Omega$). The optimal recording position for C-fiber–evoked fEPSP was at a depth of around 200 to 350 μm from the surface of the L4 lumbar enlargement. Data were acquired at a 10-kHz sampling rate. An A/D converter card (National Instruments, M-Series PCI-6221, Texas, USA) was used to digitize the data. C-fiber–evoked fEPSPs were analyzed by the WinLTP Standard Version program (WinLTP, RRID:SCR_008590, Bristol, GBR). In each experiment, the amplitude of 5 consecutive fEPSPs was averaged for overall analysis. To investigate the effects of ReaChR activation, the mean fEPSP amplitude was compared between a pre-optogenetic stimulation phase (baseline) and following light stimulation. Light stimulation (20 Hz: 45-ms light on, 5-ms light off) was applied for 30 min after a stable C-fiber–evoked fEPSP was established.

## Whole-cell recording of microglia in spinal cord slices

Mice were anesthetized with isoflurane, and coronal slices of spinal cord (350 μm) were prepared in ice-cold, sucrose-substituted artificial cerebral spinal fluid (ACSF, in mM): Sucrose 185, KCl 2.5, CaCl$_2$ 0.5, MgCl$_2$ 10, NaHCO$_3$ 26, NaH$_2$PO$_4$ 1.2, glucose 25. Slices were then incubated in normal ACSF containing (in mM) NaCl 120, KCl 2.5, CaCl$_2$ 2, MgCl$_2$ 2, NaHCO$_3$ 26, NaH$_2$PO$_4$ 1, glucose 11 for 1-h recovery. Whole-cell, patch-clamp recordings from microglia were obtained from spinal cord slices perfused with room temperature ACSF. Recordings were made using a 5-7M$\Omega$ glass pipette filled with intracellular solution, containing (in mM) K-Gluconate 135, HEPES 10, Mg-ATP 4, Na2-GTP, Phosphocreatine disodium salt hydrate 10 with pH at 7.2 and 290 to 300 mOsm. Slices were protected from light throughout the whole process of slice recovery and recordings. Microglia were held at −60 mV. In assessing a cell's current–voltage relationship in response to ReaChR activation, 2 protocols were used under voltage clamp mode: (1) a ramping protocol was used to shift the membrane voltage from the holding voltage of −60 mV to a voltage between −140 mV and 40 mV (500-ms duration); and (2) constant membrane voltage changes from −140 mV and 40 mV (500-ms duration) with a 20-mV step size. These tests were performed in both the presence and absence of red light stimulation (625 nm), which was specifically applied during the 500-ms duration of the voltage shift. In addition, current clamp mode was used to examine membrane potential shifts in response to red light stimulation. Recordings were amplified and filtered at 2 kHz by a patch-clamp amplifier (Axon Instruments, Axon 700B, California, USA), digitized (Axon Instruments, DIGIDATA 1550), and analyzed by pCLAMP software (Molecular Devices, Union City, California, USA). Data were discarded when the input resistance changed >20% during recording.

## Primary microglia cultures and IL-1β ELISA

Mixed glia cultures were prepared from P0-P2 CX3CR1[creER/+]: R26[LSL-ReaChR/+] mice and CX3CR1[creER/+] mice as a control group following an established procedure [90]. Briefly,

extracted forebrains were dissociated in 0.25% trypsin-EDTA (Thermo Fisher Scientific Gibco, cat#25200–056) and DNase I (15 unit/mL, Sigma Aldrich, cat#DN25). Cells were seeded in DMEM/F12 (Thermo Fisher Scientific Gibco, cat#11330–032) containing 10% heat-inactivated fetal bovine serum (Thermo Fisher Scientific Gibco, cat#26140–079) and 1× penicillin/streptomycin (Thermo Fisher Scientific Gibco, cat#15140–122) on 75-cm$^2$ tissue culture flasks coated with poly-D-lysine (100 μg/mL, Sigma Aldrich, cat#P6407) and incubated at 37°C in 5% $CO_2$ atmosphere. The culture medium was changed after 3 days and then 4 days until confluency (11 to 12 days in vitro). To obtain pure microglia cell, flasks of mixed glial cultures were shaken at 200 rpm for 1 h at 37°C. Floating microglia were collected by centrifugation (1,000 rpm for 10 min) and seeded at approximately $1 \times 10^5$ cells/ml on poly-D-lysine-coated dishes containing 50% old/50% new media. On the following day, culture medium was supplemented with 5 μM 4-OHT and incubated for 24 h for optimal induction of ReaChR expression in microglia. To chelate extracellular $Ca^{2+}$ during light stimulation, EGTA (Sigma, Cat# E3889, 5 mM) was used. ReaChR-expressing microglia were activated for 30 min with 625 nm, red LED light (20 Hz: 45-ms light on, 5-ms light off). The conditioned media was used to evaluate levels of IL-1β using the mouse IL-1β ELISA kit (Thermo Fisher Scientific Cat#88-7013-22, RRID: AB_2574942) following the manufacturer's protocol.

## Western blot analysis

Lumbar 4 to 5 spinal dorsal horns were collected at various time points, and protein was extracted. A total of 40 μg of protein from each group was then loaded and separated by SDS-PAGE, transferred to a PVDF membrane, blocked with 5% skim milk in TBST, and incubated overnight with primary antibodies at 4°C. Primary antibodies include rabbit anti-phospho-P38 (1:1,000; 4511S, RRID:AB_2139682, CST), rabbit anti-P38 (1:1,000; 8690T, RRID: AB_10999090, CST), rabbit anti-phospho-ERK1/2 (1:1,000; 4370S, RRID: AB_2315112, CST), rabbit anti-ERK1/2 (1:1,000; 4695S, RRID:AB_390779, CST), rabbit anti-phospho-JNK (1:1,000;4668S, RRID:AB_823588, CST), rabbit anti-JNK (1:1,000; 9252S, RRID:AB_2250373, CST), rabbit anti-phospho-NFκB P65 (1:1,000; 3033S, RRID:AB_331284, CST), rabbit anti-NFκB P65 (1:1,000; Ab32536, RRID:AB_776751, Abcam), and GAPDH (1:1,000; sc-32233, RRID:AB_627679, Santa Cruz Biotechnology). Membranes were incubated with horseradish peroxidase-conjugated goat anti-rabbit IgG (1:2,000; 111-036-045, RRID:AB_2337943, Jackson ImmunoResearch Labs, Pennsylvania, USA) and horseradish peroxidase-conjugated goat anti-mouse IgG (1:2,000; 115-035-003, RRID:AB_10015289, Jackson ImmunoResearch Labs) for 1 h at room temperature. Membranes were then treated with West Pico substrate (34078, Thermo Fisher Scientific), and chemiluminescence signal was detected with a G:BOX Chemi XRQ gel doc (Syngene, Frederick, Maryland, USA). Optical density of each band was then determined using Fiji (NIH). Results indicate expression normalized with GAPDH.

## Real-time PCR

Total RNA from isolated L4/5 ipsilateral dorsal spinal cord tissue was extracted using a standard method of TRIzol reagent (Invitrogen, cat#15596026, California, USA) [91]. cDNA synthesis was performed using iScript cDNA Synthesis kit (Bio-Rad, cat#1708890, California, USA). Real-time PCR was performed using SsoAdvanced Universal SYBR Green Supermix (Bio-Rad, cat#1725270) on LightCycler II (Roche cat#05015278001). The primer sequences of BDNF are forward: 5′-TACCTGGATGCCGCAAACA-3′ and reverse: 5′-AGTTGGCCTTTG GATACCGG-3′, IL-1β are forward: 5′-CTGTGTCTTTCCCGTGGACC-3′ and reverse: 5′-CAGCTCATATGGGTCCGACA-3′, IL-6 are forward: 5′-TTCCATCCAGTTGCCTTCTT-3′ and reverse: 5′-CAGAATTGCCATTGCACAAC-3′, TNFα are forward: 5′-GTGGAACTGG

CAGAAGAGGC-3′ and reverse: 5′-AGACAGAAGAGCGTGGTGGC-3′, CCL2 are forward: 5′-GTTGGCTCAGCCAGATGCA-3′ and reverse: 5′-AGCCTACTCATTGGGATCATCTTG-3′, GAPDH are forward: 5′-TCCATGACAACTTTGGCATTG-3′ and reverse: 5′-CAGTCTTCTGGGTGGCAGTGA-3′. The threshold cycle (Ct) of the GAPDH gene was used as a reference control to normalize the expression level of the target gene ($\Delta C_T$) to correct for experimental variation. Relative mRNA levels were calculated according to the $2^{-\Delta\Delta C_T}$ method [92].

## Drug administration

The IL-1ra (R&D, Cat# 480-RM-010, i.t., 50 ng/mL) was used to inhibit IL-1β signaling. To block pannexin, probenecid (Life Technologies, Cat# P36400, 100 mg/kg, California, USA) was intraperitoneally administered. To inhibit PKCα signaling, PKCα inhibitor peptide (Cayman Chemical, Cat#17478, i.t, 10 μM, Michigan, USA) was used. For i.t. drug administration, mice were hand restricted and injected by direct lumbar puncture between L5 and L6 vertebrae of the spine, using a 10-μL Hamilton syringe (Hamilton Bonaduz, Bonaduz, CHE) with a 31G needle. A tail flick response indicated successful insertion.

## Statistical analysis

Quantification of fluorescent immunostaining results was done with Fiji software (Fiji, RRID: SCR_002285). Pain behaviors were analyzed using 2-way ANOVA with multi-comparisons to test for main effects between groups followed by post hoc testing for significant differences. Two-group analysis utilized the Student $t$ test. Three-group analysis utilized a 1-way ANOVA design. Data are presented as mean ± SEM. All statistical analyses were performed using GraphPad Prism 8 software (GraphPad Prism 8, RRID:SCR_002798). Level of significance is indicated with $^*P < 0.05$, $^{**}P < 0.01$, $^{***}P < 0.001$, $^{****}P < 0.0001$.

## Supporting information

**S1 Data. Underlying numerical data and statistical analysis for Figs 1D, 2C, 2E, 3C, 3D, 3E, 3F, 3G, 3H, 4B, 4C, 4E, 4G, 5C, 5G, 5I, 6B, 6D, 6G, 6H, 7B, 7C and 7D and S1C, S2C, S3A, S3B, S5B, S6B, S7B, S8A, S8B, S9A, S9B, S9C, S9D, S9E, S9F, S10A, S10B, S10D, S11A, S11B, S12B Figs.**
(XLSX)

**S1 Raw Images. Original images supporting all western blot results reported in Fig 6C and S7A Fig.**
(PDF)

**S1 Fig. Expression of microglial ReaChR in CX3CR$^{creER/+}$: R26$^{LSL-ReaChR/+}$ transgenic mice.**
(A) Representative immunostaining images of rhodopsin (red) with either Iba1, GFAP, or NeuN (green) in the spinal dorsal horn of CX3CR1$^{creER/+}$: R26$^{LSL-ReaChR/+}$ mice without TM injection. Rhodopsin expression was not detected under these conditions. Scale bar, 40 μm. (B) Representative immunostaining images of rhodopsin with Iba1 in the cortex of CX3CR$^{creER/+}$: R26$^{LSL-ReaChR/+}$ mice with or without TM injection. Rhodopsin expression was co-localized with Iba1$^+$ cells in the cortex of CX3CR$^{creER/+}$: R26$^{LSL-ReaChR/+}$ mice when TM was administered but was absent without TM injection. Scale bar, 40 μm. (C) Summarized data showing the co-localization of Rhodopsin with Iba1$^+$ cells. Data are presented as mean ± SEM, $n = 4$ mice/group, $^{****}P < 0.0001$. Unpaired Student $t$ test. For data plotted in graphs, see S1 Data. ReaChR, red-activated channelrhodopsin; TM, tamoxifen.
(TIF)

**S2 Fig. Optogenetic stimulation of spinal microglia did not induce ATF3 expression in DRG.** (A) Representative immunostaining images of ATF3 in L4-5 DRGs of ReAChR mice following light stimulation. Very few ATF3$^+$ cells were observed following stimulation. Scale bar, 40 μm. (B) SNT surgery induced significant ATF3 expression within ipsilateral L4 DRG (Ipsi.) but not contralateral side (Contra.). Scale bar, 40 μm. (C) Summarized data showing the ATF3$^+$ cells. $n$ = 4 mice/group, $^{****}P < 0.0001$. Two-way ANOVA with multiple comparisons. For data plotted in graphs, see S1 Data. DRG, dorsal root ganglion; ReAChR, red-activated channelrhodopsin; SNT, spinal nerve transection.
(TIF)

**S3 Fig. Sex differences were not observed for mechanical allodynia following optogenetic stimulation of microglia.** (A, B) Measurement of mechanical (A) and thermal (B) pain hypersensitivity in male and female ReAChR mice. Results show that no difference was observed between male and female mice in regard to mechanical allodynia. However, thermal hypersensitivity was observed to last longer in female mice when compared to male mice. Data are presented as mean ± SEM, $n$ = 5 mice/group $^{****}P < 0.0001$, male vs. female. Two-way ANOVA with multiple comparisons. For data plotted in graphs, see S1 Data. ReAChR, red-activated channelrhodopsin.
(TIF)

**S4 Fig. Optogenetic stimulation of microglia induces proliferation.** (A) Timeline of experimental procedures. (B) Immunofluorescence images showing co-localization of Ki67 (red) and Iba1 (green) at 1, 3, 5, and 7 days after optogenetic stimulation compared with control group. Arrows indicate the Ki67$^+$Iba1$^+$ cells. Scale bar, 40 μm.
(TIF)

**S5 Fig. Morphological analysis of ReAChR-expressing microglia at 1 h, 7 and 9 days following optogenetic stimulation.** (A) Representative images of spinal cord microglia immunostained by Iba1 and skeletonized structure following optogenetic stimulation. Scale bar, 40 μm. (B) Summarized Sholl analysis data showing no significant change in the complexity of ReAChR microglia at 1 h, or 7 and 9 days after light stimulation compared with the control group. Data are presented as mean ± SEM, $n$ = 5 mice/group, Two-way ANOVA. For data plotted in graphs, see S1 Data. ReAChR, red-activated channelrhodopsin.
(TIF)

**S6 Fig. Optogenetic stimulation induces microglial Kv1.3 expression.** (A) Representative immunostaining images showing expression of Kv1.3 (green) in Iba1$^+$ (red) microglia at 30 min, 1 h, 1, 3, 5, and 9 days after light stimulation in the ipsilateral dorsal horn. Results indicated Kv1.3 expression increased gradually from 30 min to 3 days after light stimulation in ReAChR mice as compared to control group. Scale bar, 40 μm. (B) Summarized data showing the co-localization of Kv1.3 with Iba1$^+$ cells. Data are presented as mean ± SEM, $n$ = 4 mice/group, $^{***}P < 0.001$, $^{****}P < 0.0001$. One-way ANOVA with multi-comparisons. For data plotted in graphs, see S1 Data. ReAChR, red-activated channelrhodopsin.
(TIF)

**S7 Fig. Optogenetic stimulation of ReAChR mice induces up-regulation of p-P38 and p-Erk.** (A, B) Representative western blot images (A) and quantification data (B) showing expression of MAPK signaling molecules (P38, Erk, and JNK) and NF-$\kappa$B p-P65 in L4-5 level of the dorsal horn after optogenetic stimulation. Up-regulated expression of P-p38 and P-Erk was observed at 1 to 3 days in the ipsilateral spinal dorsal horn compared control mice after

light stimulation. However, no significant difference in P-JNK and NF-$\kappa$B p-P65 was observed. Data are presented as mean ± SEM, $n$ = 4–6 mice/group, GAPDH was used as internal control. ***$P$ < 0.001, ****$P$ < 0.0001. One-way ANOVA with multi-comparisons. For data plotted in graphs, see S1 Data. Uncropped western blot images were included in the Supporting information file. ReaChR, red-activated channelrhodopsin.
(TIF)

**S8 Fig. Inhibition of PKCα signaling alleviated microglial ReaChR-induced pain hypersensitivity.** (A, B) Light-induced mechanical (A) and thermal (B) pain hypersensitivity in ipsilateral or contralateral side of ReaChR mice after treatment of PKCα inhibitor peptide or vehicle. Data represented as mean ± SEM, $n$ = 4 mice/group. ****$P$ < 0.0001, **$P$ < 0.01, *$P$ < 0.05 vehicle vs. PKCα inhibitor. Two-way ANOVA with multi-comparisons. For data plotted in graphs, see S1 Data. ReaChR, red-activated channelrhodopsin.
(TIF)

**S9 Fig. Optogenetic stimulation increases *IL-1β* mRNA level of but BDNF release from microglia remains unaltered.** (A-E) Real-time PCR analysis data showing *mRNA* expression fold change of *IL-1β* (A), *BDNF* (B), *TNFα* (C), *IL-6* (D), *and CCL2* (E), in the L4-5 level of the dorsal horn after optogenetic stimulation. GAPDH was used as a reference control to normalize the expression level of the target gene ($\Delta C_T$) to correct for experimental variation. Relative mRNA levels were calculated according to the $2^{-\Delta\Delta C_T}$ method. (F) Pooled ELISA results showing no significant change of BDNF in culture media from primary microglia following optogenetic stimulation. *$P$ < 0.05, **$P$ < 0.01. One-way ANOVA with multi-comparisons. For data plotted in graphs, see S1 Data. IL, interleukin.
(TIF)

**S10 Fig. Exogenous IL-1β induced pain hypersensitivity and increased microglial expression of IL-1β.** (A, B) Measurement of mechanical (A) and thermal (B) pain hypersensitivity of control mice after i.t. injection of recombinant IL-1β (10 μL, 20 ng/mL). Results show enhanced mechanical and thermal hypersensitivity following administration of IL-1β. $n$ = 5 mice/group *$P$ < 0.05, **$P$ < 0.01, ****$P$ < 0.0001, Vehicle vs. IL-1β. Two-way ANOVA with multi-comparisons. (C) Representative immunostaining images of IL-1β (red) expression in Iba1$^+$ (green) cells within the spinal dorsal horn1 day after vehicle/IL-1β i.t. injection. IL-1β expression in Iba1$^+$ microglia was increased in exogenous IL-1β (i.t.) injection group, whereas no expression was observed in vehicle treated group. Scale bar, 40 μm. (D) Summarized data showing the co-localization of Kv1.3 with Iba1$^+$ cells. Data are presented as mean ± SEM, $n$ = 4 mice/group, ****$P$ < 0.0001, unpaired Student $t$ test. For data plotted in graphs, see S1 Data. IL, interleukin; i.t., intrathecal.
(TIF)

**S11 Fig. Inhibition of pannexin-1 does not attenuate microglial ReaChR-induced mechanical pain hypersensitivity.** (A) Single systemic injection of probenecid (100 mg/kg) before optic stimulation did not prevent the progressive reduction in mechanical threshold. (B) Probenecid administration at 2 days after optic stimulation did not attenuate the established mechanical allodynia following microglial optogenetic activation. Data represented as mean ± SEM, $n$ = 4 mice/group. Two-way ANOVA with multi-comparisons. For data plotted in graphs, see S1 Data. ReaChR, red-activated channelrhodopsin.
(TIF)

**S12 Fig. IL-1ra attenuates PKCα immunoreactivity in the spinal cord following optogenetic stimulation.** (A) Representative images of PKCα immunostaining (red) with NeuN

(green) in the spinal dorsal horn of ReaChR mice following i.t. injection of vehicle or IL-1ra (10 μL, 50 ng/mL). PKCα expression was not detected in NeuN$^+$ cells receiving IL-1ra 3 days after light stimulation compared with vehicle group. Scale bar, 40 μm. (B) Summarized data showing the co-localization of PKCα with NeuN$^+$ cells. Data are presented as mean ± SEM, $n = 4$ mice/group, $^{****}P < 0.0001$, unpaired Student $t$ test. For data plotted in graphs, see S1 Data. IL, interleukin; IL-1ra, IL-1 receptor antagonist; i.t., intrathecal; ReaChR, red-activated channelrhodopsin.
(TIF)

## Acknowledgments

We thank members of the Wu Lab at Mayo for insightful discussions and Dr. Li-Jun Zhou for technical advice with spinal cord dorsal horn in vivo extracellular recording.

## Author Contributions

**Conceptualization:** Min-Hee Yi, Hailong Dong, Long-Jun Wu.

**Data curation:** Min-Hee Yi, Yong U. Liu, Tingjun Chen, Yanlu Ying, Jiaying Zheng, Aastha Dheer, Dale B. Bosco.

**Formal analysis:** Min-Hee Yi, Yong U. Liu, Anthony D. Umpierre.

**Funding acquisition:** Long-Jun Wu.

**Methodology:** Min-Hee Yi, Tingjun Chen, Yanlu Ying, Long-Jun Wu.

**Project administration:** Long-Jun Wu.

**Supervision:** Long-Jun Wu.

**Writing – original draft:** Min-Hee Yi, Long-Jun Wu.

**Writing – review & editing:** Anthony D. Umpierre, Aastha Dheer, Dale B. Bosco, Hailong Dong, Long-Jun Wu.

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
