## [Editor Report · Decision Letter 0]

9 Sep 2020

Dear Dr Wu, 

Thank you for submitting your manuscript entitled "Optogenetic activation of spinal microglia triggers chronic pain in mice" for consideration as a Research Article by PLOS Biology.

Your manuscript has now been evaluated by the PLOS Biology editorial staff, as well as by an academic editor with relevant expertise, and I am writing to let you know that we would like to send your submission out for external peer review.

Please re-submit your manuscript within two working days, i.e. by Sep 11 2020 11:59PM.

Kind regards,

Gabriel Gasque, Ph.D.,

Senior Editor

PLOS Biology

---

## [Decision Letter · Decision Letter 1]

6 Oct 2020

Dear Dr Wu,

Thank you very much for submitting your manuscript "Optogenetic activation of spinal microglia triggers chronic pain in mice" for consideration as a Research Article at PLOS Biology. Your manuscript has been evaluated by the PLOS Biology editors, by an Academic Editor with relevant expertise, and by four independent reviewers. You will note that reviewer 1, Zhonghui Guan, has identified himself. 

In light of the reviews (below), we will not be able to accept the current version of the manuscript, but we would welcome re-submission of a much-revised version that takes into account the reviewers' comments. We cannot make any decision about publication until we have seen the revised manuscript and your response to the reviewers' comments. Your revised manuscript is also likely to be sent for further evaluation by the reviewers.

We expect to receive your revised manuscript within 3 months. 

**IMPORTANT - SUBMITTING YOUR REVISION**

Your revisions should address the specific points made by each reviewer. While the reviewers’ tone differed, as you will see from their detailed comments, all of them felt that the use of optogenetics to activate spinal microglial was interesting. However, there was also concern that the study lacked sufficient mechanistic insight into the effects induced by your optogenetic stimulation paradigm. This was seen as particularly important given the disconnect between the long-lasting behavioral pain effects and short duration neural effects. Therefore, for us to consider a revision, you should expand your study with substantial new experimentation to provide a comprehensive mechanistic analysis that addresses the issues raised by the reviewers, particularly reviewers 3 and 4.

Please submit the following files along with your revised manuscript:

*Re-submission Checklist*

*Published Peer Review*

*PLOS Data Policy*

*Blot and Gel Data Policy*

Sincerely,

Gabriel Gasque, Ph.D.,

Senior Editor,

ggasque@plos.org,

PLOS Biology

REVIEWS:

Reviewer's Responses to Questions

Reviewer #1, Zhonghui Guan: In this manuscript Yi et al nicely demonstrates that 30-min in vivo optogenetic activation of spinal cord microglia leads to increased microglial IL-1b production, enhanced C-fiber responses and spinal cord neuron activities, resulting in mechanical and thermal hypersensitivities that last for several days. The experiments are well designed, and the results are overall convincing. However, I have the following minor concerns.

1. As the authors point out in the Discussion, optogenetic activation may not represent the physiological function of microglia. The fact that a 30-min optogenetic stimulation results in super-long pain behaviors also suggests that the stimulation is probably unnaturally strong. As the result, the authors need to clarify in the Discussion that although optogenetic activation of microglia can lead to enhanced neuronal activity and pain behavior, its application in understanding microglia function in nature chronic pain conditions might be limited.

2. In Introduction, a citation should be added to support the claim that "Purinergic signaling can directly activate ionotropic P2X4 and P2X7 receptors that are highly calcium permeable".

3. It is unclear what tests were performed in Fig. 3E-F.

4. It would be interesting to know is optogenetic activation can induce microglia proliferation.

5. There is no control in Fig. S4C.

6. As the authors point out, there is no direct evidence suggesting the increased intracellular Ca2+ after optogenetic stimulation of microglia, they probably should tune down the function of ReaChR-induced Ca2+ in Discussion.

7. In the "Optogenetic ferrule implantation and light stimulation" part of Materials and Methods, "craniotomy" should not be the word to describe the procedure because it is the procedure to remove part of the skull bone for exposing brain. In the same section it is unclear how the authors can ensure the dura membrane to be intact.

8. In Fig. 3G, 3H, 7C, 7D and S3, is it true that the optogenetic stimulation of microglia occurred right before the baseline tests?

Reviewer #2: This manuscript examines consequences of activating spinal microglia via optogenetic methods on pain behaviors in mice. The authors show that activated microglia trigger long lasting hypersensitivity that is mediated via an IL1beta dependent activation of C-fiber inputs. The use of optogenetics to activate microglia is a very nice approach (despite the caveats stated by the authors), however, I am not so sure that the actual findings are all that novel. It has been shown by others that spinal injection of ATP activated microglia mediates a rapid pain response. It has also been shown by others that IL1beta is produced by microglia and that this causes pain responses - therefore I do not see a large conceptual advance here other than to say that the authors are able to acutely activate microglia and get a response in vivo (which is cool). This work may be better suited as a technical report.

Specific points:

I am not sure that the characterization of the ReaCHr2 current needs to be in a full main figure - it seems to me that this is mostly a control to show that the construct is functionally expressed. This could thus be moved to a supplementary figure. 

It seems to me that Fig 4 should precede Figure 3 in terms of logical flow - one would normally first demonstrate that light in facts activates the microglia, prior to doing behavioral measurements 

Some form of quantification should be provided for Fig 5F and G. The authors in fact state that they saw a significant increase in PKC expression, but I wonder whether this is semantics, or whether there was in fact quantification along with statistical analysis. How often were the experiments in Fig 5 F and G repeated?

No n values are given for Fig. 6A or fir Fig 7. This is also true for some of the supplemental data. Each legend should clearly state the numbers of repetitions.

The discussion does not mention recent work on the role of Pannexins in microglial pain signaling, which is pertinent here (see work from Trang lab in Nature Medicine and Science Advances). Does the opto-activation of microglia trigger pannexin pore formation? Does probenecid block the opto induced effects observed here? 

Reviewer #3: This is an elegantly designed and comprehensive study which reveals novel aspects on the role of spinal microglial activation, achieved via optogenetic stimulation, and its consequence on inducing pain sensitization. The authors show that a 30-minute light stimulation, which specifically depolarizes spinal microglia, triggers chronic pain hypersensitivity lasting for up to 8 days. Light stimulation leads to the adoption of an activated microglial phenotype characterized by a deramified morphology, up-regulation of cellular activation markers and the release of IL-1beta. As a central pro-inflammatory mediator, IL-1beta induces an increase in nociceptive transmission as shown by larger C-fiber-evoked field potentials which is abolished by IL-1 receptor antagonism. Remarkably, blockade of IL-1beta signaling not only prevents the enhanced nociceptive neuronal activity, but also completely abrogates pain hypersensitivity. In summary, by using a technically advanced approach this work demonstrates that a transient optogenetic depolarization of spinal microglia is a sufficient stimulus to rapidly activate microglia and trigger long-lasting pain hypersensitivity in mice via an IL-1beta-dependent mechanism.

Despite the pivotal role of IL-1beta signaling, it remains mechanistically unclear how ReaChR2 activates microglia, especially concerning the role of [Ca2+]i. ReaChR2 stimulation broadly mimics the activity of non-selective cation channels by inducing cell depolarization via non-selective cation fluxes across the membrane. However, as the authors discuss, a strong and sustained ChR2 activation may also elevate [Ca2+]i and is expected to decrease intracellular [K+]; key signaling events that are mechanistically associated with microglial activation, e.g. by regulating inflammasome assembly (see https://pubmed.ncbi.nlm.nih.gov/30755589/). Furthermore, sustained ChR2 activation is known to increase extracellular [K+] (https://pubmed.ncbi.nlm.nih.gov/31116972/), which may indirectly affect neuronal activity in addition to IL-1beta. To better understand the mechanisms by which ReaChR2 activates microglia, changes of [Ca2+]i following light stimulation should be examined (although the discussion implies that this has been tested, data is not shown), as well as the potential role of Ca2+ in promoting ReaChR2-induced release of IL-1beta (related to Fig. 6, e.g. by omitting Ca2+ from the ACSF). Likewise, potential changes in extracellular K+ should be examined after 30 minutes of ReaChR2 stimulation. Although difficult to address experimentally, the potential impact of a decreased [K+]i following ChR2 stimulation (compared to changes in membrane voltage) on inducing microglial activation and IL-1beta release should at least be discussed. 

Mechanistically, the authors show that light-evoked microglial activation rapidly enhances nociceptive transmission by augmenting C-fiber responses (Fig. 5C), while blockade of IL-1beta signaling prevents these effects and completely abrogates pain hypersensitivity (Fig. 7). However, as shown in Fig. 5C, the increase in C-fiber responsiveness is only transient and lasts only for ~ 90 min before returning to baseline after 2h. How do the authors explain the transient nature of an increased neuronal activity (i.e. C-fiber response) with the chronicity of the induced pain phenotype (mechanical allodynia and thermal hyperalgesia), which lasts for more than 7 days (Fig. 3G, H)? It seems that the initial IL-1beta dependent rise in neuronal activity acts as a trigger to induce long-lasting cellular changes (such as the observed up-regulation of PKC-alpha) which then give rise to sustained pain hypersensitivity. Additional experiments are required to gain further mechanistic insights into how PKC-alpha produces long-term changes in neuronal transmission related to the observed chronic pain phenotype.

A striking outcome of this work is the rapid onset (and longevity) of pain hypersensitivity (<1h up to 8 days) after a just 30-minute light-induced depolarization of spinal microglia (Fig. 3G, H). However, to better correlate the time course of the behavioral (and neuronal) effects with early signs of microglial changes, it would be essential to investigate this also at 30 min - 1 h after light stimulation (by examining microglial morphological changes and up-regulation of classical microglial activation markers in addition to Kv1.3, such as CD68 or IBA-1, related to Fig. 4). Moreover, microglial activation and morphology should also be investigated after the pain symptoms have recovered at 9 days post stimulation. 

Other concerns:

The resting membrane potential of patch-clamped spinal microglia in voltage clamp (by using voltage steps) shows a reversal potential of ~ -60 mV (Fig. 2C), while in current clamp mode (Fig. 2F) the cells seem to be much more depolarized at ~ -20 mV (Fig. 2F). What is the average resting membrane potential of spinal microglia and how is the difference in Vrest between the two recordings modes to be explained? In addition, what is the rationale for using an intracellular patch solution with a pH set to 7.5? (rather than 7.1–7.2, which is the physiological resting intracellular pH of cells, https://pubmed.ncbi.nlm.nih.gov/19997129/)

How do the authors explain the comparatively rapid release (<1 h) of IL-1beta in response to light stimulation in the absence of a priming stimulus, which normally is required to trigger inflammasome assembly and activation of caspase-1 (Fig. 6)?

The up-regulation of c-fos (Fig. 5F) appears to occur also at the non-treated contralateral side. Please provide more convincing immunohistochemical or other evidence to better support this claim.

Despite the recently emerging controversial views on the role of P2X4-induced expression of microglial BDNF to mediate neuropathic pain by changing neuronal chloride gradients (see e.g. https://pubmed.ncbi.nlm.nih.gov/16355225/, https://pubmed.ncbi.nlm.nih.gov/31315039/, https://pubmed.ncbi.nlm.nih.gov/24089642/), I wonder how the proposed IL-1beta-dependent mechanism fits in with the prevailing BDNF/chloride story. Did you test whether ChR2 stimulation mimics some aspects of P2X4 signaling and, apart from IL-1beta, also causes the release of other microglial mediators such as BDNF or TNF-alpha? Mechanistic parallels and differences between the P2X4/BDNF- and ChR2/IL1-beta-dependent pathways on inducing neuropathic pain should also be discussed in more detail.

Reviewer #4: This is a nice study demonstrating optogenetic activation of spinal microglia to induce pain behaviors and increases in C-fiber evoked responses. The study is well controlled and the results are of broad interest. There are several issues to be addressed. 

1. The investigation of the activation state of microglia is a bit thin. The morphological analysis is appropriate, but there are not clear links to other endpoints in this study (e.g., IL-1b). What about activation of signaling cascades, such as MAP kinases and NFkB? The authors could also consider transcriptional profiling of opto-activated microglia to provide a more comprehensive assessment of the activation state.

2. It is interesting that acute IL-1ra enduringly prevents nociceptive hypersensitivity induced by ReaChR stimulation. The data indicate that ReaChR stimulation induces acute release of IL-1b that induces secondary processes, supported by experiment showing that IL-1ra administered 3 days after light stimulation does not reduce PKCa levels in spinal neurons. This should be confirmed behaviorally. The cellular mechanism(s) should be investigated. For example, does calcium influx activate NLRP3 inflammasomes to cleave constitutive pro-IL-1b? Is autocrine signaling of IL-1b at microglia necessary to establish the persistent nociceptive hypersensitivity, or is it maintained by the neurons?

3. The sex of the mice should be included in the title and abstract.

---

## [Decision Letter · Decision Letter 2]

11 Feb 2021

Dear Dr Wu,

Thank you for submitting your revised Research Article entitled "Optogenetic activation of spinal microglia triggers chronic pain in mice" for publication in PLOS Biology. I have now obtained advice from the original reviewers and have discussed their comments with the Academic Editor. You will note that reviewer 1, Zhonghui Guan, has signed their comments.

Based on the reviews, we will probably accept this manuscript for publication, provided you satisfactorily address the remaining point raised by reviewer 4. Please also make sure to address the following data and other policy-related requests listed below my signature.

We expect to receive your revised manuscript within two weeks. 

*Published Peer Review History*

*Early Version*

Sincerely,

Gabriel Gasque, Ph.D.,

Senior Editor,

ggasque@plos.org,

PLOS Biology

ETHICS STATEMENT:

-- Please include the ID number of your protocol approved by the Institutional Animal Care and Use Committee (IACUC) at Mayo Clinic.

-- Please include the specific national or international regulations/guidelines to which your animal care and use protocol adhered. Please note that institutional or accreditation organization guidelines (such as AAALAC) do not meet this requirement.

DATA POLICY:

Note that we do not require all raw data. Rather, we ask for all individual quantitative observations that underlie the data summarized in the figures and results of your paper. For an example see here: http://www.plosbiology.org/article/info%3Adoi%2F10.1371%2Fjournal.pbio.1001908#s5

These data can be made available in one of the following forms:

Regardless of the method selected, please ensure that you provide the individual numerical values that underlie the summary data displayed in the following figure panels: Figures 1D, 2CE, 3CDEFGH, 4BCEG, 5GI, 6BDGH, 7BCD, S1C, S2C, S3AB, S5B, S6B, S7B, S8AB, S9ABCDEF, S10ABD, S11AB, and S12B.

Please also ensure that each figure legend in your manuscript includes information on where the underlying data can be found and that your supplemental data file/s has/have a legend.

Reviewer remarks:

Reviewer #1, Zhonghui Guan: I have no more concerns of the paper. 

Reviewer #2: No further comments.

Reviewer #3: All individual points were satisfactorily addressed and incorporated in the revised manuscript. I very much appreciate the additional experimental work carried out, which now yields a more comprehensive picture of the story.

Regarding my point #4 on the different magnitudes of resting potential depending on whole-cell modes: I assume that when voltage clamping the cells at -60mV, potential tonic conductances e.g. from delayed rectifier K+ channels may be present and thus shift Vm towards more negative values, while at more depolarized voltages around -20mV in the voltage follower (current clamp) mode these K+ conductances are inactivated.

Regarding the suggested more prominent role of ReaChR-induced Ca2+ influx, rather than membrane depolarization, in the discussion: I agree with this interpretation, and especially a compact and highly resistant cell type such as microglia may be significantly affected even by subtle ion (K+/Ca2+) permeabilities mediated by the inserted channelrhodopsins, and both Ca2+ influx and K+ efflux are well known signaling events required for inflammasome activation and IL-1beta release.

Reviewer #4: The authors have done an excellent job in addressing comments from all reviewers, having added substantially more data to the manuscript. 

I have only a minor comment regarding this sentence from the introduction: 

"However, it is still unclear whether direct microglial activation is sufficient to trigger pain hypersensitivity."

This has been partly addressed using DREADDs, and should be acknowledged (https://pubmed.ncbi.nlm.nih.gov/29530713/)

---

## [Editor Report · Decision Letter 3]

24 Feb 2021

Dear Dr Wu,

On behalf of my colleagues and the Academic Editor, Ben Emery, I am pleased to say that we can in principle offer to publish your Research Article "Optogenetic activation of spinal microglia triggers chronic pain in mice" in PLOS Biology, provided you address any remaining formatting and reporting issues. These will be detailed in an email that will follow this letter and that you will usually receive within 2-3 business days, during which time no action is required from you. Please note that we will not be able to formally accept your manuscript and schedule it for publication until you have made the required changes.

PRESS

Thank you again for supporting Open Access publishing. We look forward to publishing your paper in PLOS Biology. 

Sincerely, 

Gabriel Gasque, Ph.D. 

Senior Editor 

PLOS Biology